# The zinc transporter Slc30a1 (ZnT1) in macrophages plays a protective role against attenuated *Salmonella*

**Pinanong Na-Phatthalung[1,2], Shumin Sun[1], Enjun Xie[1], Jia Wang[3], Junxia Min[1]\*, Fudi Wang[1]\***

[1]The Second Affiliated Hospital, School of Public Health, State Key Laboratory of Experimental Hematology, Zhejiang University School of Medicine, Hangzhou, China; [2]The First Affiliated Hospital, Institute of Translational Medicine, Zhejiang University School of Medicine, Hangzhou, China; [3]School of Public Health, Zhengzhou University, Zhengzhou, China

**\*For correspondence:**
junxiamin@zju.edu.cn (JM);
fwang@zju.edu.cn (FW)

**Competing interest:** The authors declare that no competing interests exist.

**Abstract** The zinc transporter Slc30a1 plays an essential role in maintaining cellular zinc homeostasis. Despite this, its functional role in macrophages remains largely unknown. Here, we examine the function of Slc30a1 in host defense using mice models infected with an attenuated stain of *Salmonella enterica* Typhimurium and primary macrophages infected with the attenuated *Salmonella*. Bulk transcriptome sequencing in primary macrophages identifies Slc30a1 as a candidate in response to *Salmonella* infection. Whole-mount immunofluorescence and confocal microscopy imaging of primary macrophage and spleen from *Salmonella*-infected *Slc30a1*flag-EGFP mice demonstrate Slc30a1 expression is increased in infected macrophages with localization at the plasma membrane and in the cytosol. *Lyz2*-Cre-driven *Slc30a1* conditional knockout mice (*Slc30a1*fl/fl;*Lyz2-Cre*) exhibit increased susceptibility to *Salmonella* infection compared to control littermates. We demonstrate that Slc30a1-deficient macrophages are defective in intracellular killing, which correlated with reduced activation of nuclear factor kappa B and reduction in nitric oxide (NO) production. Notably, the model exhibits intracellular zinc accumulation, demonstrating that Slc30a1 is required for zinc export. We thus conclude that zinc export enables the efficient NO-mediated antibacterial activity of macrophages to control invading *Salmonella*.

## Editor's evaluation

The important work described in this manuscript reveals a new pathway in nutritional immunity: the zinc transporter SLC30A1 in the antimicrobial function of macrophages. Authors provide convincing evidence that zinc homeostasis promotes macrophage cell function that is not conducive to the intracellular proliferation of *Salmonella*, specifically attenuated *Salmonella*. This will be of interest to readers involved in pathogenesis, immunity, and bacteriology.

## Introduction

*Salmonella*, the gram-negative bacterium that causes Salmonellosis, is a well-characterized and relatively common foodborne disease. According to the 2015 Global Burden of Disease Study, *Salmonella* infection is one of the leading causes of diarrhea-related death, particularly among children under 5 years of age (*Troeger et al., 2017*). Severe cases of *Salmonella* septicemia are associated with a 25% mortality rate (*Feasey et al., 2012*). Based on epidemiological reports, zinc deficiency has been linked to the incidence of *Salmonella* infection (*Troeger et al., 2017*). As an essential micronutrient, zinc

is involved in a wide range of fundamental biological processes, including modulating the immune system's response to invading pathogens (*Wessels et al., 2021*). However, whether zinc plays a protective role in response to *Salmonella* infection remains unclear.

In mammalian cells, zinc homeostasis is tightly regulated by two families of zinc transporters, namely solute carrier family 30 (Slc30a, also known as ZnTs) and solute carrier family 39 (Slc39a, also known as ZIPs), which export and import cellular zinc, respectively. Zinc imbalance caused by the dysfunction of zinc transporters contributes to impaired hematopoiesis and macrophage survival (*He et al., 2023*; *Gao et al., 2017*). Slc30a1 (ZnT1) is ubiquitously expressed in mammalian cells and is localized primarily at the plasma membrane (*Nishito and Kambe, 2019*; *Golan et al., 2015*; *Guo et al., 2010*), where it enables the export of cytoplasmic zinc to the extracellular space (*Shusterman et al., 2017*; *Kambe et al., 2015*; *Kimura and Kambe, 2016*). In mice, loss of Slc30a1 causes early embryonic death (*Andrews et al., 2004*), suggesting that Slc30a1 is essential for early development and survival. Recent data provide evidence that Slc30a1 expression is induced by a wide range of pathogens, including bacteria (*Botella et al., 2011*; *Stocks et al., 2021*), fungi (*Rossi et al., 2021*), and viruses (*Moskovskich et al., 2019*).

Macrophages serve as the front line in the innate immune system by detecting and eliminating foreign pathogens. When *Salmonella* invades tissues, the bacteria are rapidly engulfed and killed by resident macrophages. This rapid engulfment and killing of *Salmonella* by macrophages result in bacterial clearance, cytokine production, and the recruitment of polymorphonuclear phagocytes (*Mastroeni et al., 2009*). Despite evidence suggesting a link between zinc regulators and macrophage function (*Gao et al., 2018*), whether and how Slc30a1 regulates macrophage function during infection remains poorly understood.

Here, we perform transcriptome sequencing analysis to identify Slc30a1 as an important host factor in *Salmonella*-infected primary macrophages. We then used *Slc30a1*$^{flag-EGFP}$ reporter mice to assess subcellular localization and cell-specific expression of Slc30a1 at the protein level. To study the role of Slc30a1 in macrophages, we generate *Slc30a1*$^{fl/fl}$;*Lyz2-Cre* (Slc30a1 cKO) mice and induce *Salmonella* infection. Lastly, we used primary macrophages differentiated from the bone marrow of Slc30a1 cKO mice to directly investigate the killing capacity and the antimicrobial response of macrophages to *Salmonella* infection. Our findings provide interesting new insights into the role of Slc30a1 in manipulating cellular zinc homeostasis to support an antimicrobial response in macrophages in response to infectious diseases.

## Results

### *Slc30a1* expression is induced in mouse macrophages upon *Salmonella* infection

We performed RNA sequencing (RNA-seq) on bone marrow-derived macrophages (BMDMs) that were isolated from wild-type C57BL/6 mice and then exposed to an attenuated stain of *Salmonella enterica* Typhimurium with a multiplicity of infection (MOI) of 1 for 2 hr, the approximate time that has been closely correlated with the initial replication of *Salmonella* in macrophages (*Helaine et al., 2010*). Our analysis revealed a total of 1074 differentially expressed genes (DEGs), defined as an absolute log$_2$ fold change >1 and an adjusted p value of <0.05 compared to control-treated BMDMs (*Figure 1A, B*). Among these 1074 genes, significantly downregulated genes were enriched in anti-inflammatory pathways, including *Gpx1*, *Smad7*, and *Tgfb3*, and significantly upregulated genes were enriched in infectious and inflammatory pathways, including *Tnf*, *Il6*, *Il1β*, *Ptgs2*, and *Nos2*, suggesting a potent antimicrobial response in infected macrophages (*Figure 1C*); as shown in a heatmap, many genes were upregulated following *Salmonella* infection (*Figure 1D*). Gene ontology (GO) analysis of the DEGs revealed that the top 50 GO terms generally contributed to proinflammatory signaling, inflammatory cytokines, and responses to molecules of bacterial origin (*Figure 1—figure supplement 1* and *Supplementary file 1*), while Kyoto Encyclopedia of Genes and Genomes (KEGG) pathway enrichment analysis of these DEGs yielded 54 pathways (p < 0.05) involved largely in infectious diseases (*Figure 1E* and *Supplementary file 2*).

Using KEGG pathway analysis, we identified mineral absorption as a predominant pathway, with upregulation of the *Atp2b1*, *Atp7a*, *Mt2*, *Slc30a1*, and *Trpm7* genes. To narrow down the list of genes, we analyzed publicly available bulk RNA-seq data (GSE67427) obtained from human monocyte-derived

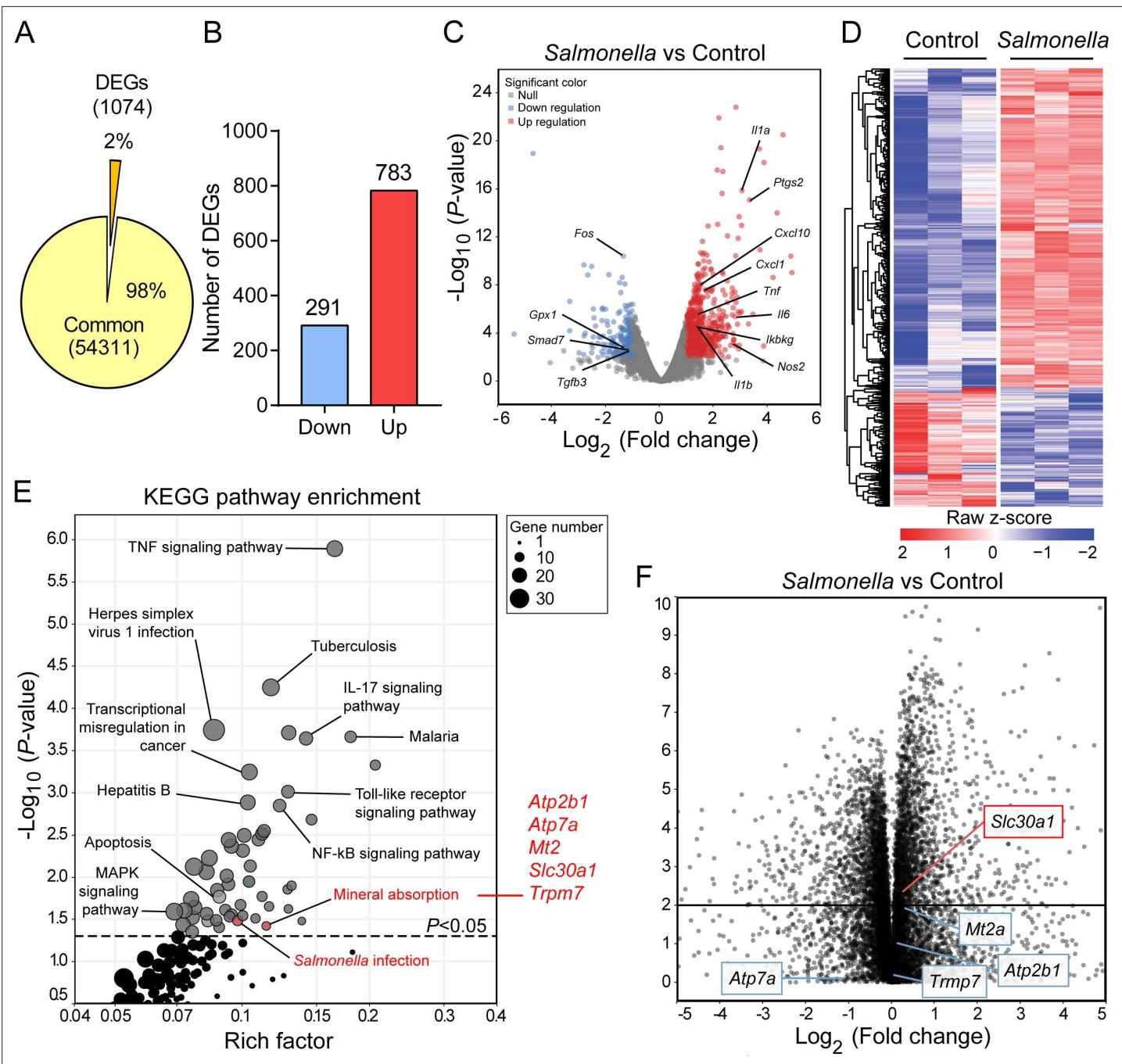

**Figure 1.** *Slc30a1* expression is upregulated in *Salmonella*-infected macrophages. (**A**) Pie chart summarizing the percentage of unchanged genes and differentially expressed genes (DEGs) in wild-type (WT) bone marrow-derived macrophages (BMDMs) 2 h after *Salmonella* infection (multiplicity of infection [MOI] = 1) (*n* = 3) using RNA sequencing (RNA-seq) analysis. (**B**) Bar graph displaying the number of DEGs including up- and downregulated genes identified by p < 0.05 and an absolute log$_2$ fold change >1; shown at the right is the total number of down- and upregulated genes. (**C**) Volcano plot displaying the fold change in gene expression and corresponding p values for *Salmonella*-infected cells versus control (uninfected) cells. (**D**) Heatmap showing the pattern of DEGs in BMDMs measured between *Salmonella*-infected cells and control cells. (**E**) Kyoto Encyclopedia of Genes and Genomes (KEGG) enrichment pathways (p < 0.05) of DEGs between *Salmonella*-infected cells and control cells. (**F**) Schematic diagram (left) illustrating the use of RNA-seq analysis on human monocyte-derived macrophages (MDMs) infected with *Salmonella* for 4 hr and volcano plot (right) of DEGs in *Salmonella*-infected MDMs based on a previously published dataset (GSE67427); the *Atp2b1*, *Atp7a*, *Mt2*, *Slc30a1*, and *Trpm7* genes are indicated.

The online version of this article includes the following figure supplement(s) for figure 1:

*Figure 1 continued on next page*

*Figure 1 continued*

**Figure supplement 1.** Gene ontology (GO) enrichment analysis of differentially expressed genes (DEGs) in bone marrow-derived macrophages (BMDMs) with or without *Salmonella* infection.

**Figure supplement 2.** The gene expression of *Slc30a1* in macrophages is induced by *Salmonella* infection.

macrophages that were infected for 4 hr with *Salmonella typhimurium* and found that the DEGs in our mouse RNA-seq dataset were also differentially regulated in human macrophages (*Figure 1F*). Importantly, *Slc30a1* was ranked as the most significantly upregulated gene and was therefore selected for further study. To validate the RNA-seq data, we then used RT-qPCR(Real-time quantitative polymerase chain reaction) to measure *Slc30a1* mRNA levels in BMDMs at various time points following exposure to attenuated *Salmonella*, lipopolysaccharide (LPS), or heat-killed *Salmonella typhimurium* (HK-ST). We found that *Slc30a1* expression peaked at 2 and 4 hr in the *Salmonella*- and LPS-treated cells, respectively, while HK-ST produced a significantly smaller response that also peaked at 2 hr (*Figure 1—figure supplement 2A*). To validate this in vitro finding in vivo, we gave C57BL/6 mice an intraperitoneal (i.p.) injection of a sublethal dose of attenuated *Salmonella* ($1 \times 10^5$ CFU per mouse); at 4 hr post-infection (4 hpi), *Slc30a1* expression was significantly higher in peritoneal macrophages compared to uninfected mice (*Figure 1—figure supplement 2B*). Together, these data show that *Slc30a1* is upregulated in macrophages during *Salmonella* infection.

## Macrophages in *Salmonella*-infected Slc30a1 reporter mice have increased levels of Slc30a1 protein

To detect the levels of Slc30a1 protein in macrophages following *Salmonella* infection, we generated a Slc30a1 reporter mouse line expressing 3xFlag-2A-EGFP-2A-CreERT2-Wpre-pA under the control of the *Slc30a1* promoter (*Figure 2A*) and studied heterozygous offspring mice (*Slc30a1*$^{flag-EGFP/+}$). BMDMs were then isolated from *Slc30a1*$^{flag-EGFP/+}$ mice and either infected with attenuated *Salmonella* (MOI = 1) or treated with ZnSO$_4$ (40 µM) for 4 hr, followed by western blot analysis using an anti-flag antibody. Compared to untreated WT and untreated *Slc30a1*$^{flag-EGFP/+}$ BMDMs, both the *Salmonella*-infected and ZnSO$_4$-treated *Slc30a1*$^{flag-EGFP/+}$ BMDMs had significantly higher levels of the reporter protein (*Figure 2—source data 1*). Given that Slc30a1 localizes primarily to the plasma membrane (*Nishito and Kambe, 2019*; *Golan et al., 2015*; *Guo et al., 2010*), we examined the localization of Slc30a1 in *Salmonella*-infected and ZnSO$_4$-treated cells using confocal microscopy. As expected, we found that ZnSO$_4$ increased the levels of the Slc30a1-flag reporter, primarily in the plasma membrane (*Figure 2C*). Interestingly, however, the Slc30a1-flag reporter was present both in the cytosol and in the plasma membrane in *Salmonella*-infected cells, suggesting that Slc30a1 may contribute to both zinc efflux and intracellular zinc movement in response to *Salmonella* infection.

To examine the in vivo expression of Slc30a1, we infected *Slc30a1*$^{flag-EGFP/+}$ mice with an i.p. injection of attenuated *Salmonella* ($1 \times 10^5$ CFU per mouse). Given that the spleen is the largest secondary lymphoid organ (*Lewis et al., 2019*) and is a major source of bacterial burden during systemic infection (*Carreno et al., 2021*), we dissected the spleen 4 hr after infection and examined Slc30a1-flag expression in splenic macrophages. Immunostaining for Slc30a1-flag and the macrophage biomarker F4/80 revealed that Slc30a1 is expressed at high levels, specifically in splenic F4/80–positive cells at the area of red pulp in infected mice, but not in uninfected mice (*Figure 2D*). Moreover, *Salmonella*-induced expression of Slc30a1 was confirmed by measuring high GFP expression in CD11b$^+$F4/80$^+$ peritoneal macrophages (*Figure 3A*) and splenic macrophages (*Figure 3B, C*), while there was no GFP expression in other splenic immune cells (*Figure 3—figure supplement 1*).

## *Lyz2*-Cre-driven *Slc30a1* conditional knockout mice have increased susceptibility to *Salmonella* infection

To examine the role of Slc30a1 in macrophages with respect to protecting against *Salmonella* infection, we generated the conditional knockout mice with a loss-of-function allele of *Slc30a1* in myeloid cells including macrophages by crossing mice carrying a floxed *Slc30a1* allele (*Slc30a1*$^{fl/fl}$) with *Lyz2*-Cre recombinase mice (*Figure 4A*); we then used the homozygous floxed mice with heterozygous Cre (*Slc30a1*$^{fl/fl}$;*Lyz2*-Cre or Slc30a1 cKO(Conditional knockout)), with homozygous floxed littermates lacking Cre (*Slc30a1*$^{fl/fl}$) serving as a control group. Loss of *Slc30a1* expression was confirmed in Slc30a1

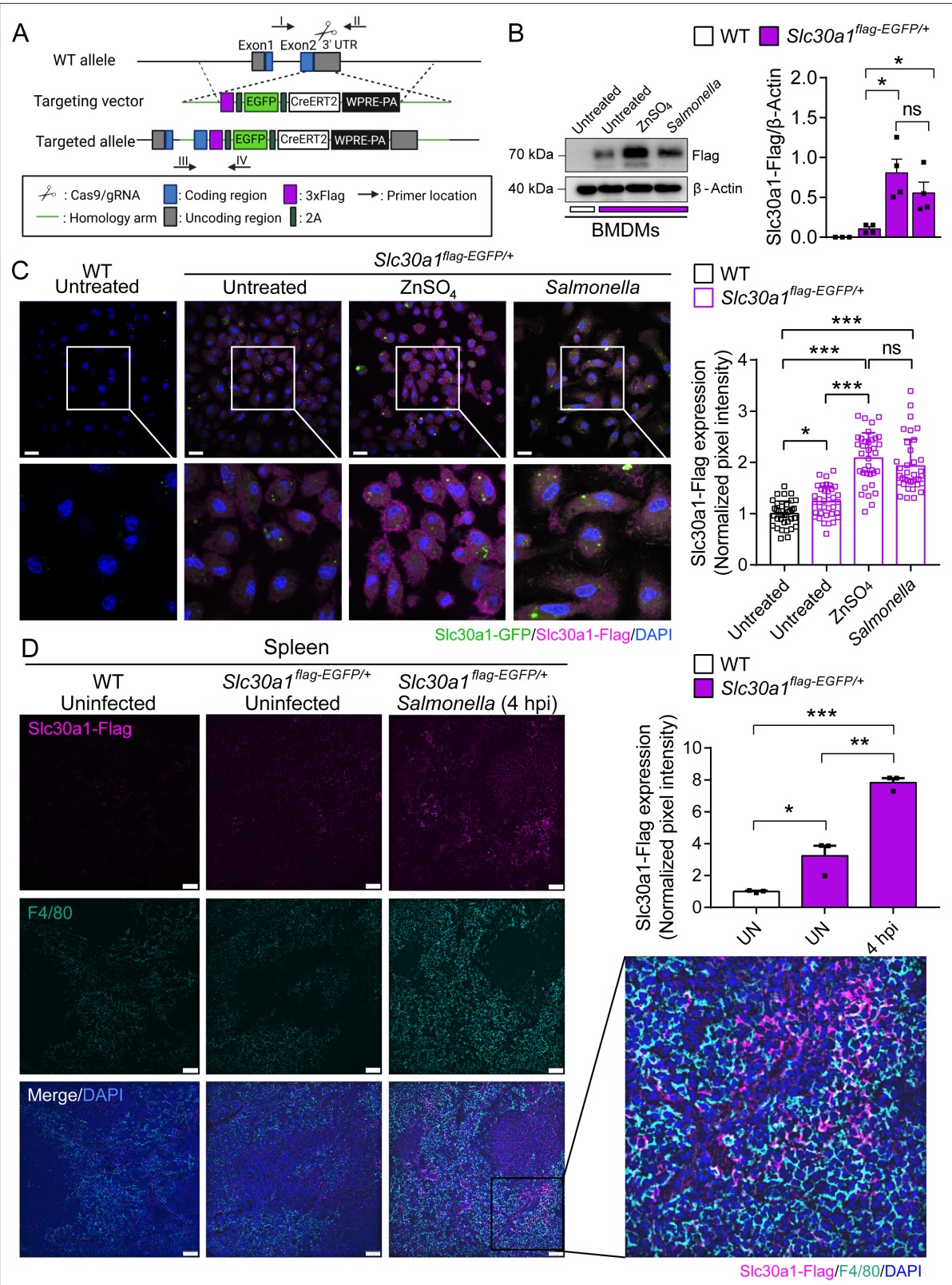

**Figure 2.** Characterization of a Slc30a1 reporter mouse with or without *Salmonella* infection. (**A**) Strategy for generating an Slc30a1 reporter mouse line expressing 3xFlag-EGFP under the control of the endogenous *Slc30a1* promotor (*Slc30a1^flag-EGFP/+^*). (**B**) Western blot analysis of Slc30a1-flag in bone marrow-derived macrophages (BMDMs) isolated from *Slc30a1^flag-EGFP/+^*mice and exposed to ZnSO₄ treatment (40 μM) or *Salmonella* (multiplicity of infection [MOI] = 1) for 4 hr. (**C**) Confocal fluorescence images of BMDMs treated as shown in B and immunostained using an anti-flag antibody

*Figure 2 continued on next page*

*Figure 2 continued*

(magenta); the nuclei were counterstained with DAPI(4',6-diamidino-2-phenylindole) (blue), and GFP(Green fluorescent protein) was visualized directly based on green fluorescence. Scale bars, 20 μm. Shown at the right is the summary of the normalized pixel intensity of Slc30a1-flag expression in BMDMs in each group. (**D**) Confocal fluorescence images of spleen samples obtained from a *Salmonella*-infected mouse at 4 hpi; also shown are samples of uninfected WT and *Slc30a1*<sup>flag-EGFP/+</sup> mice. The samples were stained with anti-flag (magenta) and anti-F4/80 (cyan), and the nuclei were counterstained with DAPI (blue). Scale bars, 50 μm. Shown at the right is the summary of the normalized pixel intensity of Slc30a1-flag expression in the spleen in each group. Data are presented as mean ± SEM(Standard Error of the Mean). p values were determined using two-tailed unpaired Student's *t*-test. *p < 0.05, **p < 0.01, ***p < 0.001, and ns, not significant.

The online version of this article includes the following source data for figure 2:

**Source data 1.** Raw images of western blot analysis for Slc30a1-flag expression.

cKO BMDMs, with a more than 95% reduction in *Slc30a1* mRNA levels compared to *Slc30a1*<sup>fl/fl</sup> cells (**Figure 4B**). Next, we injected attenuated *Salmonella* (1 × 10⁵ CFU per mouse) into male Slc30a1 cKO and *Slc30a1*<sup>fl/fl</sup> littermates; we then monitored survival for 2 weeks and sacrificed a subgroup of mice 4–48 hpi. We found that the *Salmonella*-infected Slc30a1 cKO mice had significantly lower survival compared to *Salmonella*-infected *Slc30a1*<sup>fl/fl</sup> mice (**Figure 4C**). At 24 hpi, we observed severe tissue damage in the spleen (i.e., deformation of white pulp tissue) and liver (i.e., necrotic tissue) of Slc30a1 cKO mice when compared with *Slc30a1*<sup>fl/fl</sup> mice (**Figure 4D, E**), as well as higher levels of serum TNFα, alanine aminotransferase (ALT), and aspartate aminotransferase (AST) compared to *Slc30a1*<sup>fl/fl</sup> mice (**Figure 4—figure supplement 1**). Interestingly, we also found that the *Salmonella*-infected Slc30a1 cKO mice contained a smaller proportion of CD11b⁺F4/80⁺ peritoneal macrophages compared to *Salmonella*-infected *Slc30a1*<sup>fl/fl</sup> mice (**Figure 4F**). In addition, we found significantly fewer neutrophils and monocytes in blood samples obtained at 24 hpi from infected Slc30a1 cKO mice compared to controls without interruption of other blood parameters (**Supplementary file 3**), possibly due to the reported expression of *Lyz2*-Cre in neutrophils (**Clausen et al., 1999**).

We also quantified *Salmonella* burden at 4, 24, and 48 hpi and found significantly higher numbers of bacterial cells in the peritoneal cavity, blood, spleen, and liver of infected Slc30a1 cKO mice compared to control mice, particularly at 24 hpi (**Figure 4G**), indicating severe systemic infection. Furthermore, we performed in vivo infection studies in a separate group of mice using a non-lethal dose of attenuated *Salmonella* (1 × 10⁴ CFU per mouse) and found the same pattern of tissue damage (**Figure 4—figure supplement 2A**), peritoneal macrophage recruitment (**Figure 4—figure supplement 2B**), and bacterial burden (**Figure 4—figure supplement 2C**). Together, these in vivo data provide compelling evidence that Slc30a1 in macrophages may play an important protective role against *Salmonella* infection.

## Loss of *Slc30a1* in macrophages reduces bactericidal capacity by reducing iNOS(Inducible nitric oxide synthase) and nitric oxide production

Killing intracellular microbes is a key biological function of macrophages (**Flannagan et al., 2009**). To study whether Slc30a1 mediates this function in macrophages, we isolated BMDMs from Slc30a1 cKO and *Slc30a1*<sup>fl/fl</sup> mice and exposed the cells to attenuated *Salmonella* (MOI = 10) in vitro for 24 hr. We found that Slc30a1 cKO BMDMs contained a significantly higher intracellular bacterial load compared to control cells at both 8 and 24 hr (**Figure 5A**). Moreover, examining the cells at high magnification using transmission electron microscopy revealed increased numbers of bacterial cells in the phagosomes of Slc30a1 cKO BMDMs at 24 hr compared to control cells (**Figure 5B**).

Infected macrophages typically exhibit a classically activated (i.e., M1-like) phenotype. During *Salmonella* infection, these activated macrophages produce several proinflammatory cytokines and chemokines such as IL-1β(Interleukin-1 beta), TNF, and CCL2(C-C motif chemokine ligand 2) in order to clear the pathogen via mechanisms that include cell death, the recruitment of polymorphonuclear phagocytes, and oxidative processes (**Brennan and Cookson, 2000**; **Pham et al., 2020**; **Depaolo et al., 2005**). We therefore infected BMDMs with attenuated *Salmonella* (MOI = 1) and measured the mRNA levels of genes expressing key proinflammatory cytokines (*Tnfα*, *Il1β*, and *Il6*; **Figure 5C**), chemokines (*Ccl2*, *Cxcl1*, and *Cxcl10*; **Figure 5D**), and macrophage polarization–related factors (*Arg1* and *Nos2*; **Figure 5E**). We found significantly reduced expression of proinflammatory cytokine- and chemokine-related genes in infected Slc30a1 cKO BMDMs compared to control cells. In particular,

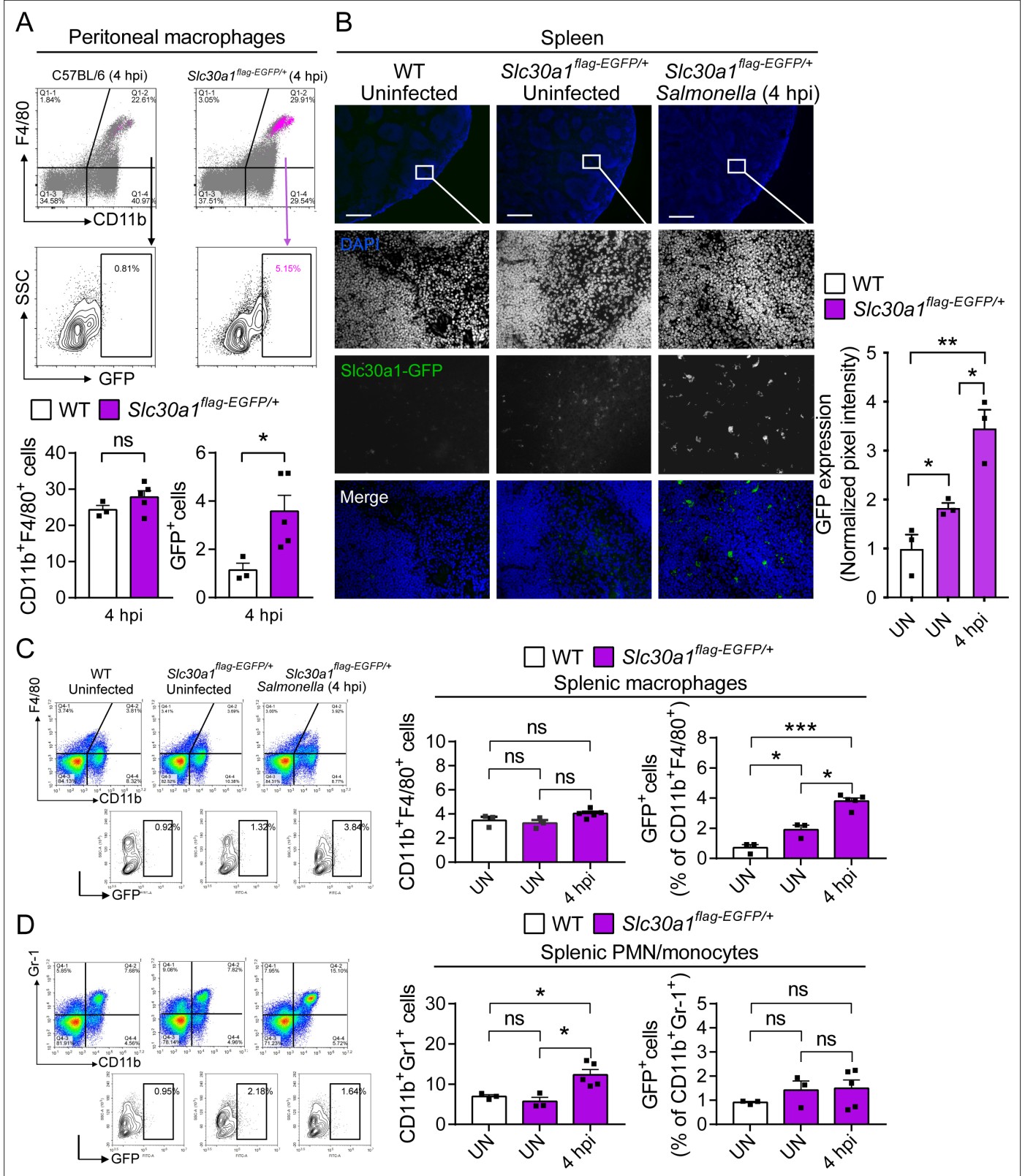

**Figure 3.** Induction of Slc30a1 expression in macrophages of a Slc30a1 reporter mouse upon *Salmonella* infection. (**A**) FACS(Fluorescence-activated cell sorting) plots of GFP expression in CD11b⁺F4/80⁺ macrophages isolated from the peritoneal cavity of *Salmonella*-infected WT mice and *Salmonella*-infected *Slc30a1*^*flag-EGFP/+*^ mice at 4 hpi (*n* = 3–5 mice/group). (**B**) Fluorescence microscopy images of spleen samples obtained from a *Salmonella*-infected mouse at 4 hpi; also shown are samples of uninfected WT and *Slc30a1*^*flag-EGFP/+*^ mice. The green region indicated the Slc30a1-GFP expression, and the

*Figure 3 continued on next page*

*Figure 3 continued*

blue region indicated the nuclei staining by DAPI. Scale bar, 500 μm. FACS plots of GFP expression in CD11b⁺F4/80⁺ splenic macrophages (**C**) and CD11b⁺Gr-1⁺ splenic PMN/monocyte (**D**) isolated from the spleens of uninfected WT mice, uninfected *Slc30a1*^flag-EGFP/+ mice, and *Salmonella*-infected *Slc30a1*^flag-EGFP/+ mice at 4 hpi (n = 3–5 mice/group). Shown at the right is the summary of the percentage of CD11b⁺F4/80⁺ splenic macrophages and CD11b⁺Gr-1⁺ splenic PMN/monocyte in each group. Data in this figure are represented as mean ± SEM. p values were determined using two-tailed unpaired Student's *t*-test. *p < 0.05, **p < 0.01, ***p < 0.001, and ns, not significant.

The online version of this article includes the following figure supplement(s) for figure 3:

**Figure supplement 1.** Proportion of splenic white blood cells in *Slc30a1*^flag-EGFP/+ after *Salmonella* infection.

we found extremely low levels of *Nos2* mRNA in infected Slc30a1 cKO BMDMs; *Nos2* encodes iNOS, a hallmark of classically activated macrophages that contributes to the host defense system by producing NO. Downregulation of the *Nos2* gene was supported by the low levels of iNOS (*Figure 5F—source data 1*) and nitrite concentrations (*Figure 5G*) in *Salmonella*-infected BMDMs. In addition, we found reduced expression of *Nos2* and a number of proinflammatory cytokine-related genes in Slc30a1 cKO BMDMs exposed to either HK-ST or LPS compared to control cells (*Figure 5—figure supplement 1A, B*). Western blot analysis of iNOS protein levels confirmed that both HK-ST and LPS (*Figure 5—figure supplement 1C, D*, *Figure 5—source data 2*) downregulate *Nos2* in Slc30a1 cKO BMDMs compared to control cells. Moreover, we measured significantly lower levels of nitrite ($NO_2^-$), a product of NO metabolism, in the cell culture supernatant of Slc30a1 cKO BMDMs 24 hr after HK-ST and LPS (*Figure 5—figure supplement 1E*) compared to control cells.

A variety of cellular signaling pathways have been reported to regulate *Nos2* expression in macrophages in response to pathogens, particularly the NF-κB (nuclear factor kappa B) pathway (*Xie et al., 1994*), the MAPK (mitogen-activated protein kinase) pathway (*Chan and Riches, 2001*), and the transcription factor STAT3 (signal transducer and activator of transcription 3) (*Ahuja et al., 2020*). We therefore measured the phosphorylated (i.e., activated) state of the principal proteins in these pathways—namely, phosphorylated p65 (in the NF-κB pathway), Erk and p38 (in the MAPK pathway), and Stat3—in Slc30a1 cKO and *Slc30a1*^fl/fl BMDMs 30 and 60 min after LPS stimulation. Western blot analysis revealed that p-p65, p-Erk, p-p38, and p-Stat3 levels were raised within 30 min in both Slc30a1 cKO and *Slc30a1*^fl/fl BMDMs; however, p-p65 levels were significantly lower in Slc30a1 cKO BMDMs at 60 min (*Figure 5—figure supplement 1F*, *Figure 5—source data 2*). Similarly, *Salmonella* infection (MOI = 1) caused lower levels of p-p65 protein in Slc30a1 cKO BMDMs at 60 min compared to infected *Slc30a1*^fl/fl cells (*Figure 5H—source data 1*), as well as reduced mRNA levels of several genes upstream of the NF-κB pathway, including *p65*, *MyD88*, *Ikkα*, and *Ikkβ* (*Figure 5I*). Notably, these *Salmonella*-induced changes in the expression of these key NF-κB signaling molecules were prevented by treating cells with the membrane-permeable zinc chelator TPEN (*N, N,N′,N′*-tetrakis-(2-pyridyl-methyl)-ethylenediamine (TPEN)) (*Figure 5J—source data 1*, *Figure 5K*). These results suggest that Slc30a1 in macrophages protects against *Salmonella* infection by regulating intracellular zinc.

## Altered zinc distribution in *Salmonella*-infected *Lyz2*-Cre-driven *Slc30a1* knockout mice

Next, we examined whether Slc30a1 in macrophages regulates systemic zinc homeostasis by measuring zinc concentration in the serum, spleen, and liver of Slc30a1 cKO and *Slc30a1*^fl/fl mice infected with attenuated *Salmonella* ($1 \times 10^5$ CFU per mouse) for 24 hr using inductively coupled plasma mass spectrometry (ICP-MS); we also measured other minerals, including iron, copper, magnesium, calcium, and manganese (*Figure 6—figure supplement 1A–C*). We found that compared to *Salmonella*-infected control mice, *Salmonella*-infected Slc30a1 cKO mice had significantly higher levels of zinc in the serum and spleen but no difference in the liver (*Figure 6A*), consistent with the previous finding that during systemic infection serum zinc is redistributed to the liver in order to increase monocyte maturation, boost the immune system, and restrict the delivery of nutrient metals to pathogens (*Alker and Haase, 2018*). In addition, elevated serum zinc may impair the function of immune cells, leading to increased susceptibility to *Salmonella* infection. Moreover, the spleen serves to filter the blood filter and is a hub for monocytes (*Swirski et al., 2009*); thus, the high zinc content in the spleen may be due to circulating high zinc–containing macrophages, a notion supported by our finding of increased mRNA

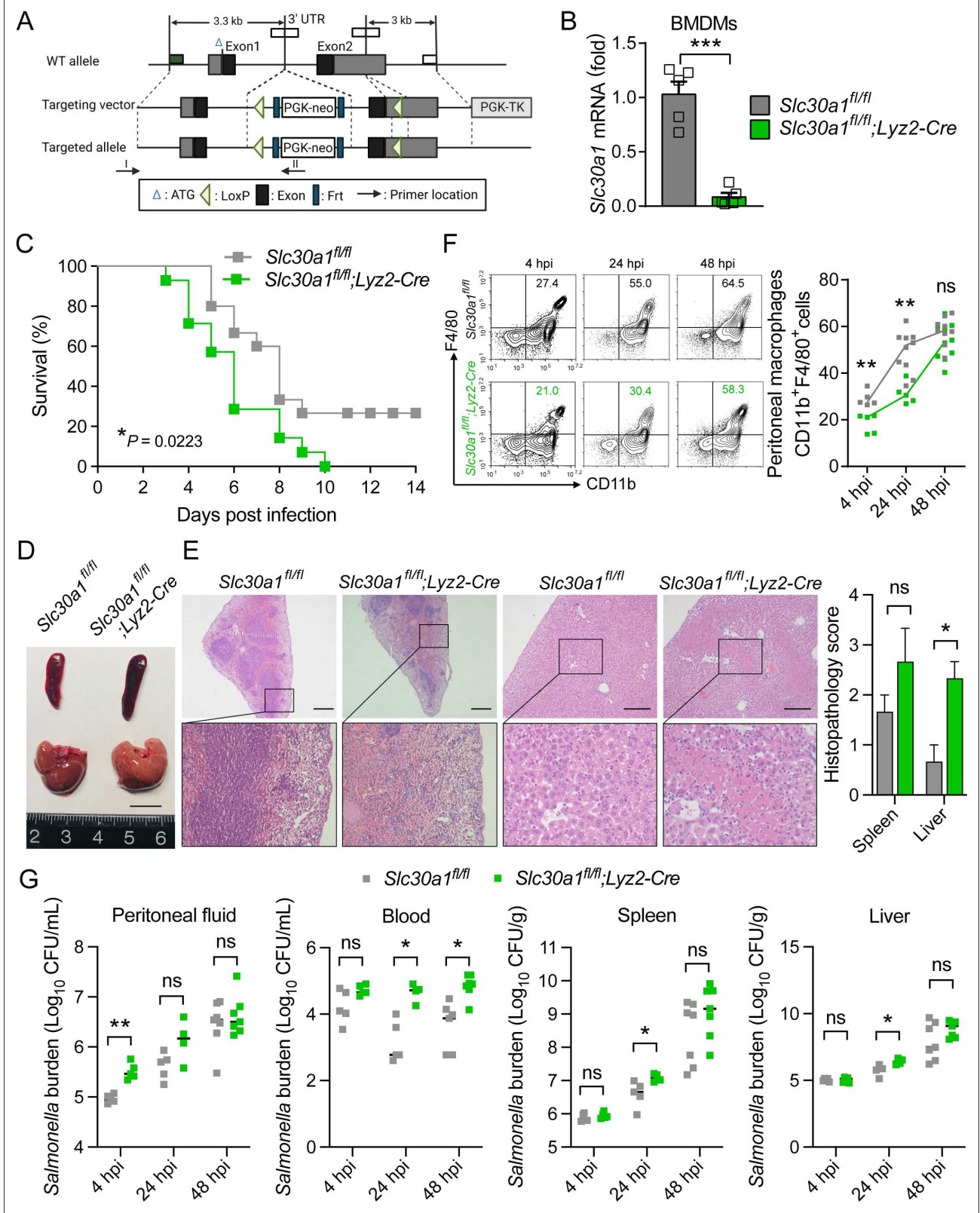

**Figure 4.** *Lyz2*-Cre-driven *Slc30a1* conditional knockout mice are highly susceptible to *Salmonella* infection. (**A**) Strategy for generating floxed *Slc30a1* mice. Crossing this mouse with a heterozygous *Lyz2*-Cre mouse produces *Slc30a1^fl/fl^;Lyz2-Cre* and *Slc30a1^fl/fl^* littermates. (**B**) RT-qPCR analysis of relative *Slc30a1* mRNA levels in bone marrow-derived macrophages (BMDMs) isolated from *Slc30a1^fl/fl^;Lyz2-Cre* and *Slc30a1^fl/fl^* mice (*n* = 5). (**C**) Kaplan–Meier survival curve of *Salmonella*-infected mice (*n* = 14–15 mice/group). (**D, E**) Gross and hematoxylin and eosin (H&E)-stained images of spleen and

*Figure 4 continued on next page*

*Figure 4 continued*

liver obtained from *Salmonella*-infected mice at 24 hpi. Scale bars, 1 cm, 5 mm and 10 mm, respectively. (**F**) FACS plots (left) and summary (right) of CD11b⁺F4/80⁺ peritoneal macrophages obtained from *Salmonella*-infected mice at 4, 24, and 48 hpi ($n$ = 5–10 mice/group). (**G**) Summary of *Salmonella* CFUs measured in the peritoneal fluid, blood, spleen, and liver of *Salmonella*-infected mice at 4, 24, and 48 hpi ($n$ = 3–5 mice/group). Data in this figure are represented as mean ± SEM. p values of survival in D were determined using Log-rank test, in B and F–K using two-tailed unpaired Student's $t$-test. *$p < 0.05$, **$p < 0.01$, ***$p < 0.001$, and ns, not significant.

The online version of this article includes the following figure supplement(s) for figure 4:

**Figure supplement 1.** Elevated levels of proinflammatory cytokine and injury markers in the serum of *Slc30a1^{fl/fl};Lyz2-Cre* mice.

**Figure supplement 2.** *Salmonella* infection leads to significantly decreased macrophages and increased tissue bacterial burden in *Slc30a1^{fl/fl};Lyz2-Cre* mice.

---

levels of *Mt1*, which encodes the principal splenic zinc reservoir protein metallothionein 1, in the spleen of *Salmonella*-infected Slc30a1 cKO mice, but not in the liver (**Figure 6B**).

Next, we used the fluorescent zinc probe FluoZin-3 to quantify the accumulation of intracellular zinc in BMDMs isolated from Slc30a1 cKO and control mice following ZnSO₄ treatment. Consistent with the well-documented role of Slc30a1 in zinc resistance (**Palmiter, 2004**), we found that Slc30a1 cKO BMDMs were significantly more sensitive to ZnSO₄-induced toxicity compared to control cells (**Figure 6—figure supplement 2**), and ZnSO₄ treatment (100 µM) caused a significantly higher FluoZin-3 signal in Slc30a1 cKO BMDMs compared to control cells (**Figure 6C, D**).

Previous studies showed that cytosolic-free zinc increases in macrophages following exposure to LPS, *Escherichia coli*, and Salmonella (**Haase et al., 2008**; **Brieger et al., 2013**, **Wu et al., 2017**). We, therefore, measured intracellular free zinc in *Salmonella*-infected Slc30a1 cKO and control BMDMs. We found that 30 min after *Salmonella* infection (MOI = 1), FluoZin-3 fluorescence was significantly higher in Slc30a1 cKO cells; similar results were obtained when the cells were exposed to H₂O₂ (1 mM), LPS (1 µg/ml), or HK-ST (MOI = 100) (**Figure 6E**). In addition, we measured higher levels of *Mt1* mRNA in Slc30a1 cKO BMDMs compared to *Slc30a1^{fl/fl}* cells, even in uninfected cells (**Figure 6F**). Based on these results, we hypothesize that the loss of Slc30a1 leads to an abnormal increase in intracellular zinc during *Salmonella* infection and increases the expression of *Mt1*, possibly to compensate for the excessive intracellular zinc by increasing Mt1-mediated zinc storage (**Figure 6G**).

## Discussion

Here, we provide evidence that *Slc30a1* expression in macrophages is required for host resistance against pathogens, as illustrated schematically in **Figure 7**. Based on this model, *Slc30a1*-deficient macrophages have increased *Salmonella* burden due to reduced iNOS and NO formation via reduced NF-κB signaling. This effect is due primarily to the inhibitory effects of high intracellular zinc, reflected by the compensatory upregulation of the zinc-binding protein Mt1.

The cellular zinc exporter Slc30a1 is a key regulator of cellular zinc homeostasis (**Kambe et al., 2015**; **Liuzzi and Cousins, 2004**; **Lichten and Cousins, 2009**). Previous in vitro studies have shown that Slc30a1 in macrophages mediates the intracellular killing of invading *Mycobacterium tuberculosis* and *E. coli* by increasing zinc levels within phagosomes, thereby promoting bacterial clearance (**Botella et al., 2011**; **Stocks et al., 2021**). Here, we provide the first in vivo evidence that Slc30a1 in macrophages plays a key role in host protection against *Salmonella* infection. We show that mice lacking Slc30a1 in macrophages are highly susceptible to *Salmonella* infection.

The killing capacity of macrophages has been studied extensively, and it mediates the host defense by ingesting and then destroying invading microbes (**Flannagan et al., 2009**). Using BMDMs isolated from our *Lyz2*-Cre-driven *Slc30a1* conditional knockout mice, we show that these macrophages have reduced pathogen-killing capacity via reduced NF-κB signaling, decreasing iNOS and NO production. Interestingly, we found that treating these cells with the zinc chelator TPEN improves NF-κB signaling, suggesting that changes in cellular zinc levels likely contribute to their reduced antimicrobial function. Wu et al. previously reported that zinc can suppress iNOS/NO production by inhibiting NF-κB in *Salmonella*-infected macrophages (**Wu et al., 2017**). Moreover, recent studies have shown that zinc can reduce NF-κB activation in macrophages (**Haase et al., 2008**; **Brieger et al., 2013**; **Wu et al., 2017**; **Jeon et al., 2000**). With respect to the underlying mechanism, studies have shown that zinc inhibits the NF-κB pathway by reducing the activity of recombinant IKKα and IKKβ by blocking IκBα

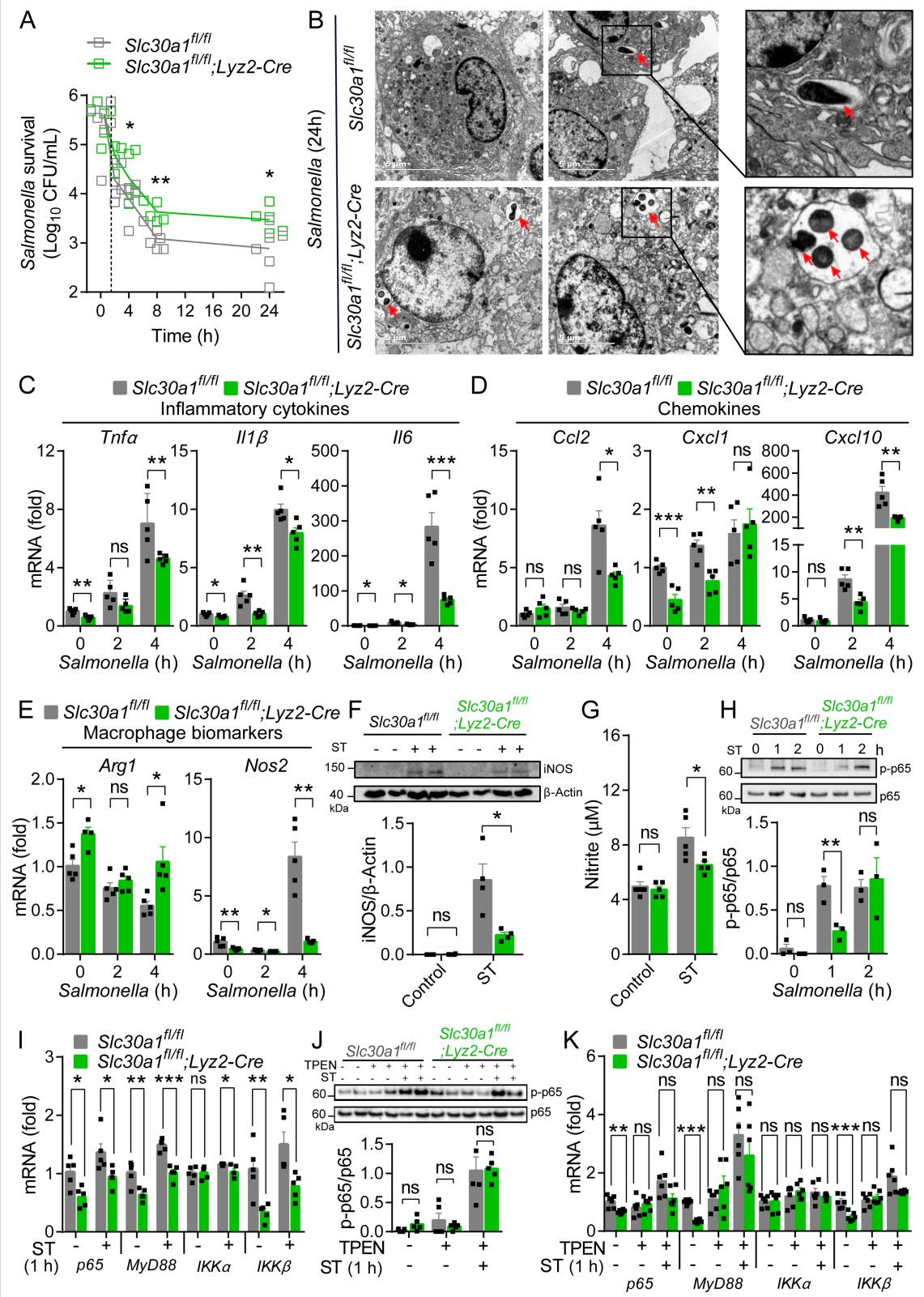

**Figure 5.** Slc30a1 is required for iNOS/nitric oxide (NO) production and intracellular pathogen-killing capacity of macrophages in response to *Salmonella* infection. (**A**) Time course of the intracellular pathogen-killing capacity of *Salmonella*-infected *Slc30a1^fl/fl^;Lyz2-Cre* and *Slc30a1^fl/fl^* bone marrow-derived macrophages (BMDMs) measured in colony-forming units per ml (*n* = 5). (**B**) Transmission electron microscopy images of *Salmonella*-infected BMDMs at 24 hpi. Red arrows indicate bacterial-containing phagosomes, and the insets show magnified images of bacterial engulfment. Scale

*Figure 5 continued on next page*

*Figure 5 continued*

bars, 5 µm. (**C–E**) RT-qPCR analysis of mRNAs encoding inflammatory cytokines (*Tnfα*, *Il1β*, and *Il6*), chemokines (*Ccr2*, *Ccl2*, *Cxcl1*, and *Cxcl10*) and macrophage biomarkers (*Nos2* and *Arg1*) in BMDMs measured at the indicated times after *Salmonella* infection (multiplicity of infection [MOI] = 1) (*n* = 5). (**F**) Western blot analysis and summary of iNOS protein in *Slc30a1^fl/fl^;Lyz2-Cre* and *Slc30a1^fl/fl^* BMDMs either untreated or stimulated with *Salmonella* for 24 hr (*n* = 3). (**G**) Summary of nitrite concentration measured in the cell culture supernatant of BMDMs either untreated or stimulated with *Salmonella* for 24 hr (*n* = 5). (**H**) Western blot analysis and summary of p-p65 and p65 measured at the indicated times in *Salmonella*-infected BMDMs (*n* = 3). (**I**) RT-qPCR analysis of mRNA levels of genes involved in nuclear factor kappa B (NF-$\kappa$B) signaling (*p65*, *MyD88*, *Ikkα*, and *Ikkβ*) measured in BMDMs 60 min after *Salmonella* infection (*n* = 5). (**J**) Western blot analysis and summary of p-p65 and p65 measured in uninfected and *Salmonella*-infected BMDMs either with or without *N,N,N',N'*-tetrakis-(2-pyridyl-methyl)-ethylenediamine (TPEN) (4 µM) for 60 min (*n* = 4). (**K**) RT-qPCR analysis of *p65*, *MyD88*, *Ikkα*, and *Ikkβ* mRNA in uninfected and *Salmonella*-infected BMDMs either with or without TPEN for 60 min (*n* = 6). Data in this figure are represented as mean ± SEM. p values were determined using two-tailed unpaired Student's *t*-test. *$p < 0.05$, **$p < 0.01$, ***$p < 0.001$, and ns, not significant.

The online version of this article includes the following source data and figure supplement(s) for figure 5:

**Source data 1.** Raw images of western blot analysis for iNOS and p65 expression.

**Source data 2.** Raw images of western blot analysis for iNOS and related-cellular signaling proteins.

**Figure supplement 1.** Loss of *Slc30a1* in the murine macrophages reduces iNOS and nitric oxide (NO) production by inhibiting the nuclear factor kappa B (NF-$\kappa$B) pathway.

phosphorylation and degradation (*Jeon et al., 2000*; *Zhou et al., 2004*; *Liu et al., 2013*). Consistent with this notion, our finding of the role of Slc30a1 in regulating the macrophage response to *Salmonella* may help explain how the genetic deletion of *Slc30a1*, specifically in this professional phagocyte leads to the observed disease outcome in our mouse model.

An interesting finding from our in vivo experiments is that our *Lyz2*-Cre-driven *Slc30a1* conditional knockout mice fail to redistribute zinc levels in the serum and spleen following *Salmonella* infection. In general, serum zinc is rapidly taken up by the liver as part of the host defense process during systemic infection (*Alker and Haase, 2018*). In mice, defects in this process—for example, due to high zinc supplementation—have negative consequences for neutrophil extracellular traps (*Kuźmicka et al., 2020*). In infants, consuming a full-fat diet of powdered cow's milk containing excessive zinc content can lead to a decreased phagocytic capacity of monocytes (*Schlesinger et al., 1993*). In adults, excessive consumption of zinc supplements can impair both the migration and bactericidal capacity of polymorphonuclear leukocytes (*Chandra, 1984*). This is consistent with our finding that a high zinc serum level in our *Lyz2*-Cre-driven *Slc30a1* conditional knockout mice affects the percentage of neutrophils and monocytes and the migration of macrophages, thereby reducing bacterial clearance and increasing their susceptibility to *Salmonella* infection. Furthermore, nutrient-dense conditions such as excessive serum zinc can be beneficial to the growth of *Salmonella*. Previous studies reported that a high-affinity $Zn^{2+}$ uptake system ZnuABC of *Salmonella* protects the bacterial cells from nitrosative stress in macrophages (*Fitzsimmons et al., 2018*). Zinc rich also induces peptide deformylase, an essential protein contributing to *Salmonella* maturation that can be inhibited by NO (*Singhal and Fang, 2020*).

Our measurements of intracellular zinc levels revealed high zinc concentrations in Slc30a1-deficient macrophages. We also detected an upregulation of *Mt1*, possibly indicating a cellular response to intracellular zinc saturation. Mt1 is a low molecular weight cysteine-rich protein with a high affinity for divalent metals that is essential for buffering surplus zinc to maintain zinc homeostasis (*Krężel and Maret, 2017*). Smith et al. showed that the level of *Mt1* expression was positively correlated with intracellular free zinc concentration (*Smith et al., 2008*). In this respect, *Mt1* overexpression may serve as zinc storage for reducing the side effects of excessive cytosolic-free zinc due to Slc30a1 deletion.

Although we found that genetically deleting *Slc30a1* in macrophages reduces the killing capacity by suppressing iNOS and nitrite production through zinc-medicated NF-κB activation, the underlying mechanism that connects NO inactivation and excessive intracellular zinc remains uncertain. In this regard, additional experiments will be needed to investigate whether the antimicrobial function of Slc30a1-deficient macrophages is restricted to the defect of NO using selective inhibitors of iNOS. Furthermore, although we focused our study on BMDMs, macrophages may respond differently to infection in the context of the full immune system in vivo. Thus, generating a mouse line in which both *Slc30a1* and *Nos2* are knocked out in macrophages (*Slc30a1^fl/fl^Nos2^fl/fl^;Lyz2-Cre*) may provide insight into underlying mechanisms behind failed bacterial clearance that affects disease outcome.

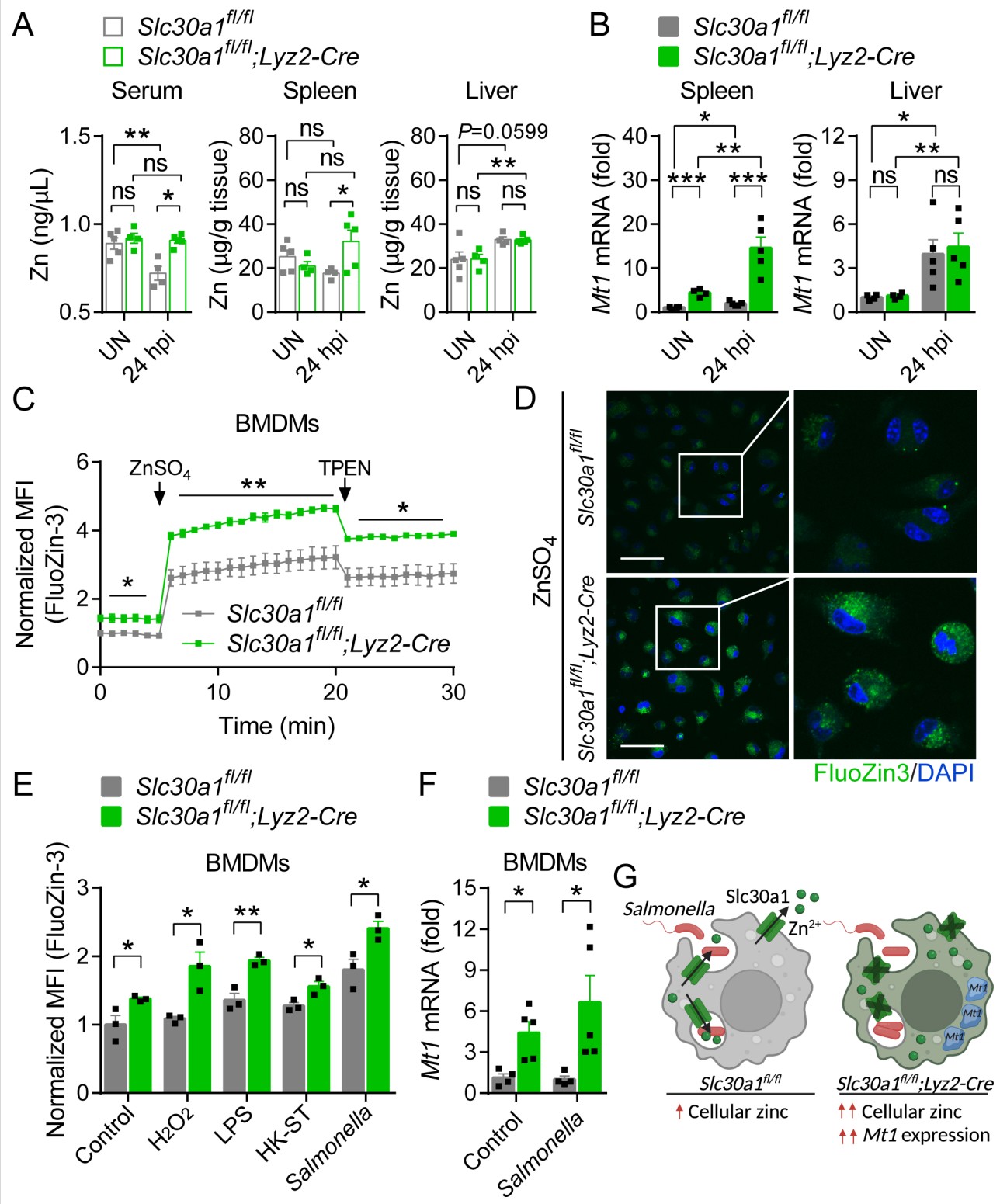

**Figure 6.** Loss of *Slc30a1* in macrophages causes intracellular zinc accumulation. (**A**) Summary of zinc (Zn) content measured in the serum, spleen, and liver of uninfected (UN) and *Salmonella*-infected mice at 24 hpi (*n* = 4–5 mice/group). (**B**) RT-qPCR analysis of *Mt1* mRNA in the spleen and liver of the indicated mice. (**C**) Time course of normalized FluoZin-3 mean fluorescence intensity (MFI) measured in bone marrow-derived macrophages (BMDMs); where indicated, $ZnSO_4$ (100 μM) and *N, N,N′,N′*-tetrakis-(2-pyridyl-methyl)-ethylenediamine (TPEN) (4 μM) were applied to the cells. (**D**) Confocal fluorescence images of BMDMs stained with FluoZin-3 (green) after treatment with $ZnSO_4$ for 15 min; the nuclei were counterstained with DAPI (blue).

*Figure 6 continued on next page*

*Figure 6 continued*

Scale bars, 50 μm. (**E**) Summary of normalized FluoZin-3 MFI measured in BMDMs 30 min after application of H₂O₂ (1 mM), lipopolysaccharide (LPS; 1 μg/ml), heat-killed *Salmonella typhimurium* (HK-ST) (multiplicity of infection [MOI] = 100), or *Salmonella* (MOI = 10) (*n* = 3). (**F**) RT-qPCR analysis of *Mt1* mRNA in uninfected and *Salmonella*-infected BMDMs (*n* = 5). (**G**) Model showing the predicted effects of the loss of Slc30a1 on cellular zinc trafficking and intracellular zinc accumulation in BMDMs in response to *Salmonella* infection. Data in this figure are represented as mean ± SEM. p values were determined using two-tailed unpaired Student's *t*-test. *p < 0.05, **p < 0.01, ***p < 0.001, and ns, not significant.

The online version of this article includes the following figure supplement(s) for figure 6:

**Figure supplement 1.** Except for Zn, *Lyz2*-Cre-mediated genetic deletion of *Slc30a1* in mice does not affect the content of other trace minerals.

**Figure supplement 2.** *Slc30a1* deficiency increase sensitivity to zinc stress.

In conclusion, we report that Slc30a1 of macrophages plays a protective role against *Salmonella* infection by regulating iNOS and NO activities, which are essential for bacterial clearance through NF-κB signaling. These findings underscore the notion that cellular zinc regulated by zinc transporter Slc30a1 is important for host defenses by maintaining an intracellular killing capacity of macrophages. However, the precise mechanism of how Slc30a1 drives NO-mediated antibacterial activity remains misty, and further studies are needed to explore this aspect.

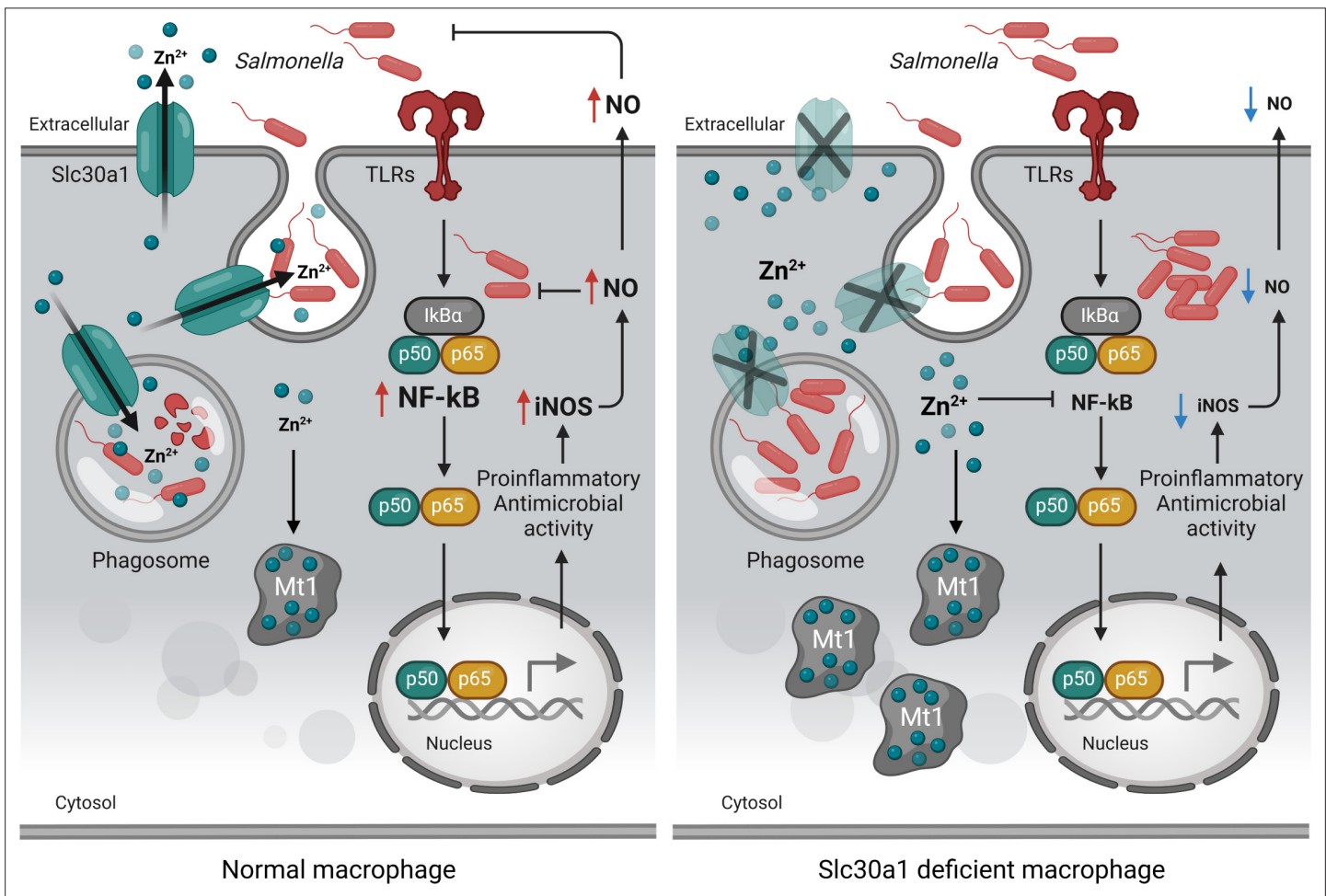

**Figure 7.** Putative protective function of Slc30a1 in macrophages during *Salmonella* infection. Under normal conditions, Slc30a1 is upregulated in response to *Salmonella* infection, generating a short-term decrease in cytosolic zinc concentration and increasing zinc toxicity in *Salmonella*-containing phagosomes. Loss of Slc30a1 leads to an accumulation of intracellular zinc, thereby upregulating *Mt1* overexpression and reducing iNOS and nitric oxide (NO) production via reduced nuclear factor kappa B (NF-κB) signaling, reducing the cell's bacterial clearance capacity.

## Materials and methods

### Mouse strains

*Slc30a1*<sup>flag-EGFP/+</sup> mice were generated by Shanghai Biomodel Organism Science & Technology Development Co Ltd. The donor vector contains the 5′ homologous arm, 3xFlag-2A-EGFP-2A-CreERT2-Wpre-pA, and the 3′ homologous arm. In the 3′ UTR of the *Slc30a1* locus, the 3xFlag-2A-EGFP-2A-CreERT2-Wpre-pA cassette was inserted downstream of exon 2. *Slc30a1* floxed mice were generated by Shanghai Biomodel Organism Science & Technology Development Co Ltd. The floxed *Slc30a1* allele contains a single loxP site upstream of exon 2 and a single loxP site neo cassette downstream of exon 2. Heterozygous *Slc30a1* floxed mice (*Slc30a1*<sup>fl/+</sup>) were backcrossed to the C57BL/6 background for more than 5 generations and then crossed with *Lyz2*-Cre-expressing mice (The Jackson Laboratory, 004781) to generate *Lyz2*-Cre-driven *Slc30a1* conditional knockout (Slc30a1 cKO) mice. All mice were genotyped using genomic PCR. Cre-negative floxed mice (*Slc30a1*<sup>fl/fl</sup>) were used as the control group. Under standard conditions, an 8-week-old male Slc30a1 cKO presents with no obvious phenotype (data not shown). Wild-type C75BL/6 mice were purchased from Shanghai SLRC Laboratory Animal Co, Ltd. Eight-week-old male mice were used in this study. All animals were fed a standard chow diet with access to drinking water and were housed at constant temperature (23°C) under a 12:12 hr light/dark cycle in specific pathogen-free conditions. All in vivo experiments were conducted in accordance with the National Institutes of Health guidelines and were approved by the Institutional Animal Care and Use Committee at Zhejiang University.

### DNA isolation and genotyping

Genomic DNA of mouse tissue biopsies was extracted using the Tissue & Blood DNA Extraction kit-250prep (Zhejiang Easy-Do Biotech Co, Ltd, #DR0301250) in accordance with the manufacturer's instructions. DNA samples were quantified using a Nanodrop 2000 spectrophotometer (Thermo Fisher Scientific). PCR amplification was conducted using 50 ng DNA per reaction, 2xTaq PCR StarMix (GenStar, #A012-101), and specific primers (*Figure 8A*) in a T100 Thermal Cycler (Bio-Rad, #10878655). Finally, the PCR products were separated by agarose gel electrophoresis to determine the genotypes (*Figure 8B, C*, *Figure 8—source data 1*).

### Bacterial preparation

The experiments were performed with attenuated *S. enterica* subsp. *enterica* serovar Typhimurium strain MeganVac1 (Accession: CP112994.1). The bacterial stain was grown in tryptic soy broth medium (Solarbio Life Science, #T8880) overnight at 37°C under sterile conditions. Bacterial inoculum was compared to a 0.5 McFarland turbidity standard (approximately $1 \times 10^8$ CFU/ml) and adjusted to $10^5$ colony-forming units per ml (CFU/ml) with sterile phosphate-buffered saline (PBS). The bacterial count in the original suspension was verified using a plate counting method. HK-ST was prepared by incubating the bacterial inoculum in a 60°C water bath for 2 hr, and a bacterial viable count was performed to confirm ≥99.99% reduction in viability.

### Primary cell culture

BMDMs were differentiated from bone marrow cells obtained from 8-week-old male mice (*Figure 9*). In brief, bone marrow cells from the femur and tibia were isolated by flushing with sterile PBS (Corning, #21-040-CV) and filtered through a 40-μm nylon membrane filter (Falcon, #350). The resulting cell suspension was centrifuged at $300 \times g$ for 5 min, and the cell pellet was resuspended in RPMI 1640 (Corning, #21-040-CVR) containing 1% (vol/vol) penicillin–streptomycin (HyClone, #SV30010), 20% (vol/vol) fetal bovine serum (FBS) (Gibco, #10270-106), and 30% L929 conditioned medium. BMDMs were then differentiated for 7 days under 37°C in humidified air containing 5% $CO_2$, refreshing the differentiation medium every 2 days. These BMDMs were then transferred to RPMI 1640 supplemented with 1% penicillin–streptomycin and 10% FBS at 37°C and used the next day. We found no difference in BDMD differentiation between the genotypes used in this study.

### RNA-seq and data analysis

BMDMs from biological triplicates of C75BL/6 mice were infected with *Salmonella* for 2 hr, and total RNA was extracted for RNA-seq. RNA-seq analysis was performed by LC Sciences (Hangzhou, China).

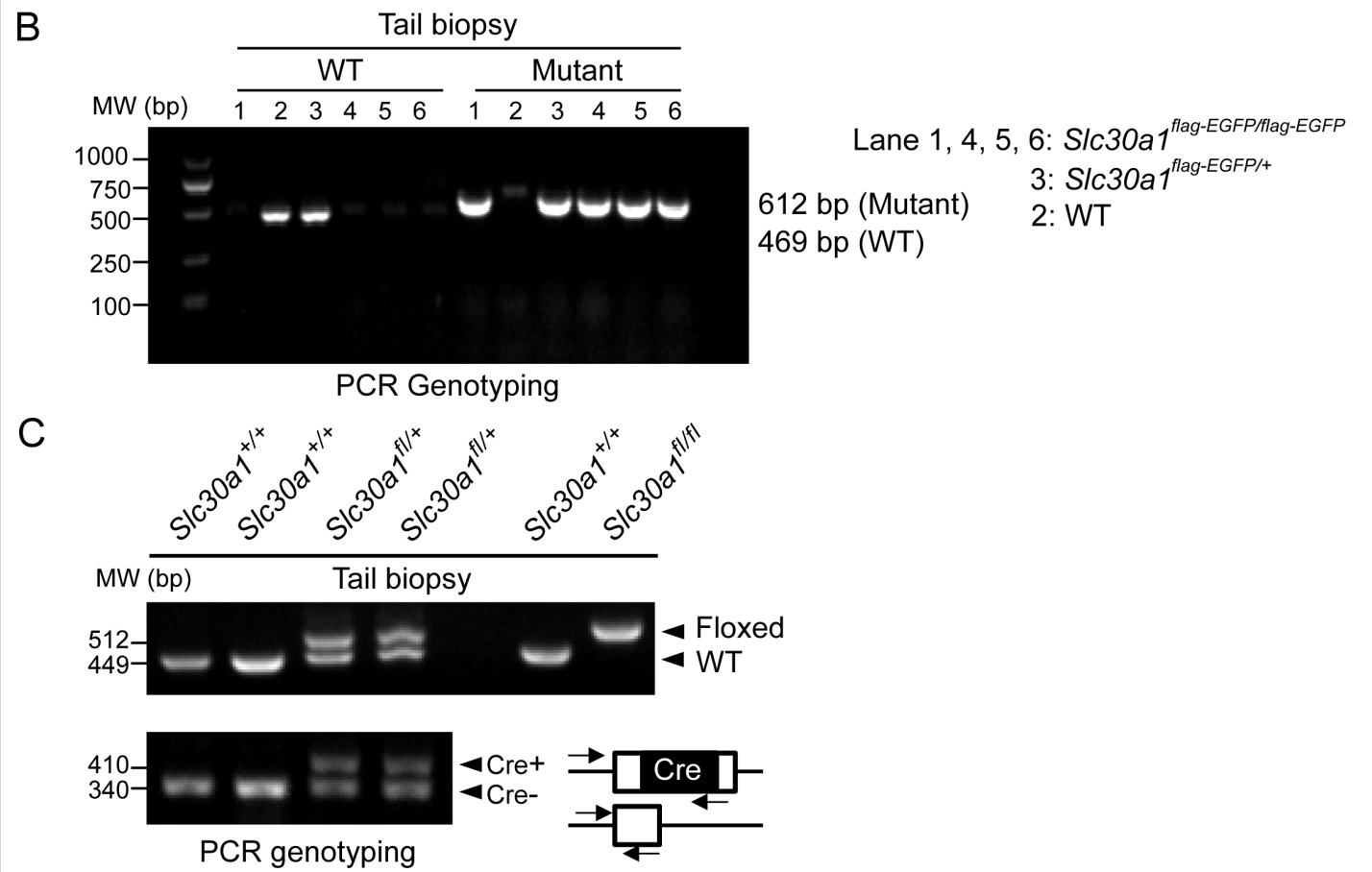

**Figure 8.** PCR-based genotyping. (**A**) List of primers used for routine genotyping. PCR was performed using genomic DNA from mouse tail biopsies. (**B**) Genotyping of *Slc30a1*-3xflag-EGFP (*Slc30a1*$^{flag-EGFP/+}$) mouse line. The PCR fragment length for the wild-type (WT) Slc30a1 allele is 612 and 469 bp for the mutant allele, respectively. (**C**) The conditional knockout (*Slc30a1*$^{fl/fl}$;*Lyz2-Cre*) mice were recognized by genomic PCR rendering one band of flox/flox allele (512 bp) with two bands of *Lyz2*-Cre heterozygous (410 and 340 bp). Each lane represents one individual mice.

The online version of this article includes the following source data for figure 8:

**Source data 1.** Raw images of PCR genotype analysis.

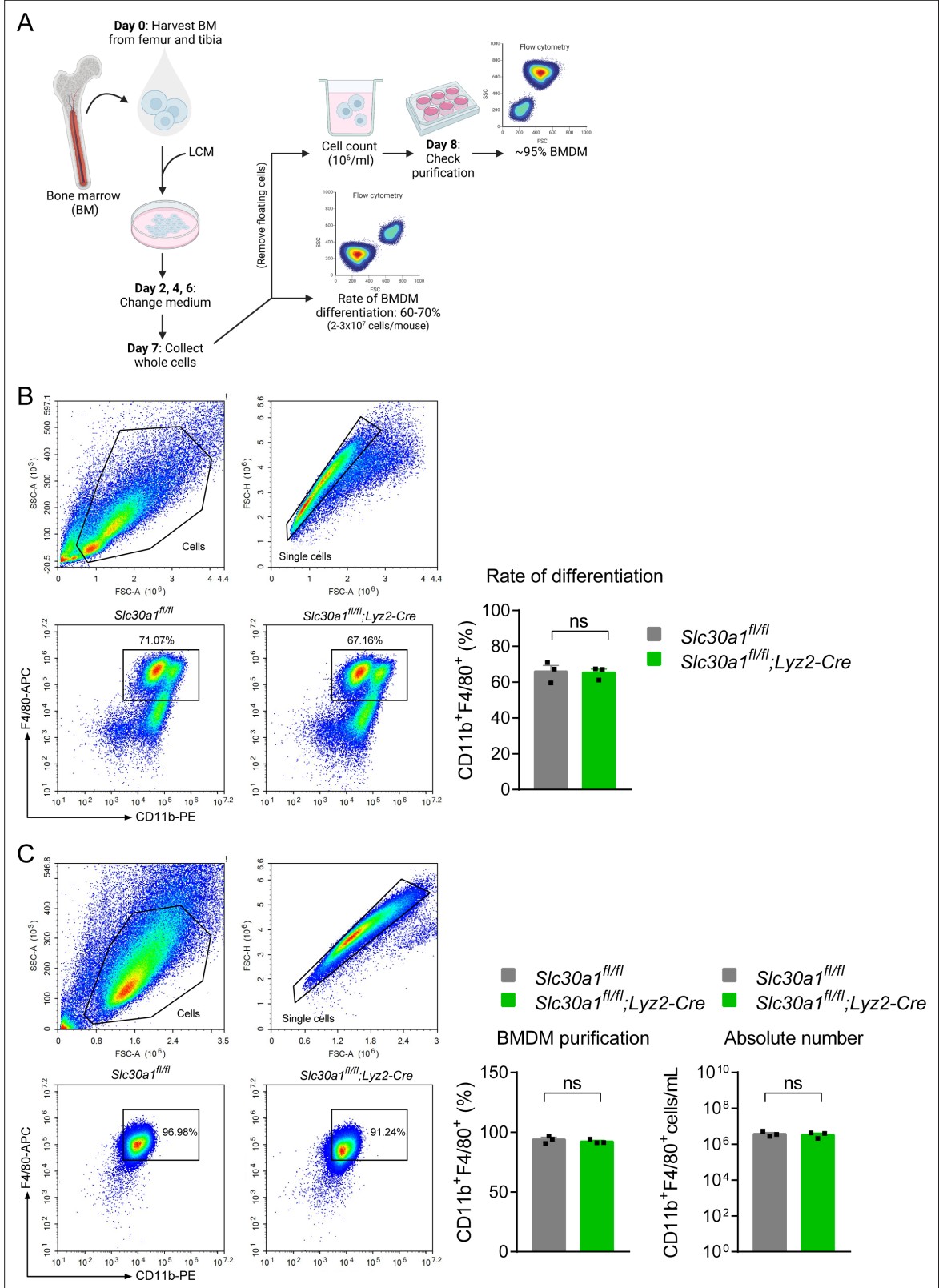

**Figure 9.** *Lyz2*-Cre-mediated genetic deletion of *Slc30a1* in mice does not affect the differentiation of bone marrow-derived macrophages (BMDMs). (**A**) Schematic illustration depicting the strategy used to measure the BMDM differentiation procedure. Bone marrow (BM) cells were harvested from the femur and tibia and cultured in L929 cell-conditioned medium (LCM) for 7 days. (**B**) Flow cytometry analysis and summary of the percent BMDM differentiation (CD11b⁺F4/80⁺) from BM samples obtained from the indicated mice measured at day 7 (*n* = 3). (**C**) Percent BMDM purification and

*Figure 9 continued on next page*

absolute number of BMDMs measured after removing floating cells and measuring cell concentration. Data in this figure are represented as mean ± SEM. p values were determined using two-tailed unpaired Student's *t*-test. ns, not significant.

GO and KEGG enrichment analyses were performed using the DAVID database. Heatmaps were generated using the OmicStudio tool (https://www.omicstudio.cn/tool). Volcano plots and bar plots of the GO and KEGG pathways were generated using the Tableau (2019.3. Ink) software desktop system.

## Immunofluorescence

BMDMs grown on glass coverslips in 6-well plates were exposed to $ZnSO_4$ (40 μM) or infected with *Salmonella* (MOI = 1) for 4 hr. The cells were then fixed in 4% paraformaldehyde (PFA) for 15 min, permeabilized in 0.2% Triton X-100 (Sigma, #T9284) for 10 min, and then blocked in 5% (wt/vol) BSA(Bovine serum albumin) (Sigma, #A6003) for 30 min at room temperature. The fixed cells were then incubated in monoclonal anti-FLAG M2-Peroxidase (HRP) antibody (Sigma, #A8592) overnight at 4°C, followed by fluorophore-conjugated secondary antibody, Alexa Fluor 647 secondary antibody (Thermo Fisher Scientific, #A32728) for 1 hr at room temperature in the dark. The nuclei were stained with 3 μM DAPI (4',6-diamidino-2-phenylindole; BioLegend, #422801) for 5 min; after gently rinsing three times with PBS, the glass coverslips were mounted cell-side down on clean plain glass slides with 80% glycerol (Sigma, #G5516) and imaged using a CSU-W1 confocal microscope (Olympus).

## *Salmonella* infection in vivo

Eight-week-old male mice were injected intraperitoneally (i.p.) with 200 μl PBS containing *Salmonella* at a final dose of $1 \times 10^5$ CFU per mouse or sterile PBS as uninfected groups. At the indicated time points, the mice were euthanized, and peritoneal cavity cells were subsequently collected to determine the peritoneal macrophage population. Blood samples were obtained via cardiac puncture, and the spleen and liver were dissected. The organs were either immediately placed on ice in sterile PBS to count bacterial burden or immersed in 4% PFA for later histological analysis. A portion of the blood was kept in serum collection tubes and allowed to be clotted for 2 hr at room temperature; clear serum was then obtained by centrifugation at 1000 rpm for 10 min. For survival experiments, after infection with *Salmonella,* the mice were monitored daily for 2 weeks.

## Bacterial burden in tissues

Approximately 100–200 mg of liver and spleen samples were homogenized in sterile PBS. A 100 μl of tissue homogenate, peritoneal cavity fluid, and blood were serially diluted 10-fold in PBS and plated on TSA(Tryptic soy agar) agar (Solarbio life science, #T8650). The plates were incubated at 37°C overnight, and viable bacteria colonies were counted as CFU/ml or CFU per gram of tissue (CFU/g).

## Serum ALT and AST measurements

The levels of serum AST and ALT from mice infected with or without *Salmonella* for 24 hr were measured using an alanine aminotransferase assay kit and aspartate aminotransferase assay kit (Shen-SuoYouFu) according to the manufacturer's protocol.

## ELISA

Cytokine levels of TNFα were quantified in serum obtained from mice infected with and without *Salmonella* for 24 hr by the mouse/rat TNF-A Valukine ELISA kit (R&D Systems, #VAL609) following the product protocol.

## Histopathology

Spleen and liver samples were immersed in 4% PFA solution for 24 hr and then embedded in paraffin. The tissue samples were then sectioned and stained with hematoxylin and eosin using standard protocols. The sections were examined using bright-field microscopy (Nikon Eclipse Ni-U). Three adjacent sections of each sample were quantified using ImageJ software (National Institutes of Health).

## Flow cytometry

Cell suspensions ($1 \times 10^5$ cells/ml) were immunostained using specific anti-mouse antibodies in the dark at 4°C for 30 min. The following fluorochrome-conjugated antibodies were used in this study (all from BioLegend): CD11b-Pacific Blue (clone M1/70, #101224), CD11b-PE (clone M1/70, #1101208), F4/80-APC (clone BM8, #123116), Gr-1-APC (clone RB6-8C5, #108412), CD45-APC/Cyanine7 (clone 30-F11, #103116), CD45R/B220-Pacific Blue (clone RA3-6B2, #103227), and CD3-PerCP/Cyanine 5.5 (clone 17A2, #100218). All antibodies were used at a dilution of 1:200. Live versus dead cells were identified based on DAPI staining. Cell sorting was performed using the NovoCyte flow cytometer (ACEA Biosciences Inc).

## Trace mineral analysis using ICP-MS

Metal content was measured using ICP-MS. Tissue samples (200 mg) were digested in 4 ml of EMSURE ISO nitric acid 65% solution (Merck, #1.00456.2508) using a MARS 6 microwave extraction system. In brief, the samples were heated from room temperature to 200°C in a microwave oven (1000 W, 2,450 MHz) for up to 20 min. The microwave power was maintained for an additional 30 min, followed by a cooling-down period of 15 min. The samples were then completely dried at 110°C for 3 hr. The digested samples were diluted to a final volume of 5 ml in high-purity deionized water obtained using a Milli-Q Integral 10 purification system (EMD Millipore, #C205110). The resulting samples were subjected to metal analysis using an Agilent 7700x ICP-MS equipped with an Agilent ASX 520 auto-sampler.

## Gene expression analysis using RT-qPCR

Total RNA was extracted from the samples using TransZol Up (TransGen Biotech, #ET111-01), and 1 µg of total RNA was used as a template to synthesize cDNA using the HiScript II 1st Strand cDNA Synthesis Kit (Yeasen Biotech, #11123ES60) in a ProFlex PCR System (Life technologies). Each RT-qPCR reaction consisted of 5 µl of SYBR Green PCR Master Mix (Bimake, #B21202), 2 µl of forward and reverse primers (1 µM), 0.2 µl of cDNA, and sufficient RNase-free water to yield a final volume of 10 µl. RT-qPCR was performed in duplicate for each sample using the LightCycler 480 Real-Time PCR System (F. Hoffmann-La Roche, Ltd) with the following conditions: initial denaturation at 95°C for 3 min, 40 cycles of amplification at 95°C for 15 s, 60°C for 30 s, and 72°C for 30 s, followed by denaturation at 95°C for 30 s, and annealing-extension at 40°C for 30 s. Data were analyzed using the $2^{-\Delta Ct}$ method (*Pfaffl, 2001*) to calculate the relative target gene expression normalized to the endogenous housekeeping gene *Gapdh* (glyceraldehyde-3-phosphate dehydrogenase). The oligonucleotides used in this study are provided in *Supplementary file 4*.

## Bacterial killing assay

BMDMs were plated at $2.5 \times 10^5$ cells/well in a 12-well plate and allowed to adhere overnight. The cells were then infected with *Salmonella* (MOI = 10) for 30 min by centrifuging the plate at 1000 rpm for 10 min and incubating for an additional 20 min. The plate was then rinsed twice with PBS containing 100 µg/ml gentamicin (Sigma, #345814) to remove the remaining extracellular bacteria. The cells were then incubated in a fresh medium containing 10 µg/ml gentamicin to prevent the growth of extracellular bacteria. Where indicated, infected cells were lysed with 0.1% Triton X-100, and serial dilutions of the cell lysates were plated on TSA agar, followed by incubation at 37°C overnight. The number of colonies appearing on the agar plate was counted in order to quantify the surviving bacteria. Intracellular killing of macrophages was verified after 24 hr infection using transmission electron microscopy. BMDMs were harvested and subjected to routine processing, post-fixed, embedded, sectioned, and mounted at the Center for Cryo-Electron Microscopy (CCEM), Zhejiang University. Finally, thin sections (~70 nm) were examined in an FEI Tecnai 10 (100 kV) transmission electron microscope.

## NO measurement

Nitric oxide (NO) was determined through the formation of nitrite ($NO_2^-$), the primary, stable metabolite of NO. Extracellular nitrite released from BMDMs ($1 \times 10^5$ cells/well) stimulated with either HK-ST (MOI = 10) or 100 ng/ml LPS (Sigma, #L7895) was measured using a colorimetric assay based on the Griess reaction (*Green et al., 1982*). At the indicated time points, cell culture supernatants were collected and placed into a fresh 96-well plate. The supernatants were then mixed with freshly

prepared Griess solution consisting of 0.1% *N*-1-naphthyl ethylenediamine dihydrochloride (Sigma, #N9125) and 1% sulfanilamide (Sigma, #V900220) in 5% phosphoric acid (Sigma, #P5811) at a ratio of 1:1. The plate was incubated at room temperature for 10 min in the dark. Absorbance at 540 nm was then measured using a microplate reader (Biotek Eon, Gen5), and nitrite production was calculated using a $NaNO_2$ calibration curve.

## Cell viability

Cell viability of BMDMs under high zinc conditions was measured using the plate-based colorimetric tetrazolium salt assay. First, BMDMs were plated in 96-well plates at $2 \times 10^4$ cells/well. The next day, twofold serial dilutions of $ZnSO_4$ (Sigma, #Z0251) were added to the culture media, and the cells were incubated at 37°C for 24 hr. Next, the cell culture medium was carefully removed and replaced with 100 µl of fresh medium containing 5 mg/ml thiazolyl blue tetrazolium blue (Sigma, #M2128). The cells were incubated for an additional 4 hr, and then the cell culture media was discarded. Blue crystals catalyzed by the mitochondrial enzyme succinate dehydrogenase were solubilized in 100% DMSO(Dimethyl sulfoxide), and the intensity was measured colorimetrically at 570 nm.

## Intracellular zinc measurement

A cell-based microplate assay was used to measure intracellular free zinc ions ($Zn^{2+}$) using the cell-permeable fluorescent probe FluoZin-3 AM (Invitrogen, #F24195). BMDMs were plated into 96-well clear-bottom black polystyrene microplates (Corning, #3603) at $1 \times 10^5$ cells/well and allowed to attach onto the bottom surface overnight. The cells were then incubated in culture media containing 2 µM FluoZin-3 AM for 30 min in the dark. After washing twice with PBS, the cells were incubated in fresh PBS for an additional 10 min to allow de-esterification. To detect cellular zinc accumulation, 100 µM $ZnSO_4$ (Sigma, #Z0251) and/or 4 µM TPEN (Sigma, #P4413) was added to the cells, and FluoZin-3 fluorescence was measured at 1-min intervals for 30 min at 37°C using a SynergyMx M5 fluorescence microplate reader (Molecular Devices) with 485 nm excitation and 535 nm emission (*Brieger et al., 2013*). The levels of intracellular zinc in response to pathogens and $H_2O_2$ were measured after 30 min stimulation. Images of FluoZin-3 fluorescence were acquired using a Nikon A1R confocal microscopy and analyzed using NIS-Elements Viewer imaging software, version 4.50.

## Protein quantification using western blot analysis

Proteins from BMDMs were extracted using RIPA buffer (Solarbio, #R0020) containing Pierce protease inhibitor (Thermo Fisher Scientific, #A32965) and phosphatase inhibitor (Roche, #04906845001). Total protein content was quantified using the Bradford dye-binding method (Sigma, #B6916). A total of 50 µg of cellular proteins were mixed with 5× loading buffer and boiled for 5 min. Equal amounts of prepared proteins were electrophoresed in 10% SDS–PAGE(sodium dodecyl sulfate-polyacrylamide gel electrophoresis) gel at 100 V. After gel electrophoresis, separated proteins were transferred onto polyvinylidene difluoride membrane (Bio-Rad, #1620177) at 300 mA for 90 min using transfer buffer containing 25 mM Tris, 192 mM glycine, and 20% methanol. The membranes were then blocked in 5% (wt/vol) skim milk in Tris-buffered saline containing 0.05% Tween-20 (TBST) for 90 min at room temperature. The membranes were then washed three times with TBST for 10 min each and then probed using the following primary antibodies overnight at 4°C (1:1000 dilution): iNOS antibody (#39898S), NF-κB p65 (D14E12) XP rabbit mAb (#8242S), or phosphor-NF-κB p65 (Ser536) (93H1) rabbit mAb (#3033) (all from Cell Signaling Technology). A monoclonal anti-FLAG M2-Peroxidase (HRP) antibody was used to detect the flag-tagged fusion protein. The following day, the membranes were washed with TBST and incubated with a secondary antibody, HRP-conjugated goat anti-rabbit IgG (H+L) (ABclonal, #AS014) diluted in 5% milk/TBST (1:2000) at room temperature for 2 hr. The membranes were then washed in TBST three times for 10 min each to remove excessive antibodies and developed using Pierce ECL Western Blotting Substrate (Thermo Fisher Scientific, #32106). The bands were quantified using the ChemiDoc Touch Imaging System (Bio-Rad).

## Statistical analysis

Statistical significance between the two groups was determined using a two-tailed, unpaired Student *t*-test (for comparing two groups). Data are presented as mean ± SEM. For comparison of survival curves, a Log-rank (Mantel–Cox) test was performed. The sample size for each statistical analysis is

provided in the figure legends. Data were analyzed and generated using GraphPad Prism version 7.04 (GraphPad Software Inc, La Jolla, CA, USA). For all studies, p values below 0.05 were considered to be statistically significant. *p < 0.05; **p < 0.01; ***p < 0.001; ns, not significant.

## Acknowledgements

This work was supported by the National Natural Science Foundation of China (32330047, 31930057 to FW and 31970689 to JM).

---

## Additional information

### Funding

| Funder | Grant reference number | Author |
| --- | --- | --- |
| National Natural Science Foundation of China | 32330047 | Fudi Wang |
| National Natural Science Foundation of China | 31930057 | Fudi Wang |
| National Natural Science Foundation of China | 31970689 | Junxia Min |

The funders had no role in study design, data collection, and interpretation, or the decision to submit the work for publication.

### Author contributions

Pinanong Na-Phatthalung, Conceptualization, Data curation, Formal analysis, Investigation, Methodology, Writing - original draft; Shumin Sun, Enjun Xie, Jia Wang, Data curation, Investigation, Methodology; Junxia Min, Fudi Wang, Conceptualization, Resources, Supervision, Funding acquisition, Validation, Visualization, Project administration, Writing - review and editing

### Author ORCIDs

Pinanong Na-Phatthalung ⓘ http://orcid.org/0000-0002-8162-4645
Fudi Wang ⓘ https://orcid.org/0000-0001-8730-0003

### Ethics

All in vivo experiments were conducted in strict accordance with the recommendations in the Guide for the Care and Use of Laboratory Animals of the National Institutes of Health. All animal experiments were approved by the Institutional Animal Care and Use Committee, Zhejiang University. Committee protocol #AIRB-2023-1479 at the institution.

### Decision letter and Author response

Decision letter https://doi.org/10.7554/eLife.89509.sa1
Author response https://doi.org/10.7554/eLife.89509.sa2

---

## Additional files

### Supplementary files

• MDAR checklist

• Supplementary file 1. Top 50 enriched GO terms for the DEGs in C57BL/6 BMDMs infected with Salmonella versus uninfected cells.

• Supplementary file 2. KEGG enrichment pathways of the DEGs in C57BL/6 BMDMs infected with Salmonella versus uninfected cells.

• Supplementary file 3. Summary of blood test results for uninfected and Salmonella-infected Slc30a1$^{fl/fl}$ and Slc30a1$^{fl/fl}$;Lyz2-Cre mice.

• Supplementary file 4. List of primer pairs used for RT-qPCR analysis.

## Data availability

The RNA-seq datasets used to generate figure 1A–F have been deposited in NCBI Gene Expression Omnibus ID GSE239543 and GSE67427, respectively.

The following dataset was generated:

| Author(s) | Year | Dataset title | Dataset URL | Database and Identifier |
|---|---|---|---|---|
| Na-Phatthalung P, Wang F, Min J | 2024 | RNA-seq data from mouse BMDMs infected with *Salmonella enterica* serovar Typhimurium | https://www.ncbi.nlm.nih.gov/geo/query/acc.cgi?acc=GSE239543 | NCBI Gene Expression Omnibus, GSE239543 |

The following previously published dataset was used:

| Author(s) | Year | Dataset title | Dataset URL | Database and Identifier |
|---|---|---|---|---|
| Blischak JD, Tailleux L, Mitrano A, Barreiro LB, Gilad Y | 2015 | Mycobacterial infection induces a specific human innate immune response | https://www.ncbi.nlm.nih.gov/geo/query/acc.cgi?acc=GSE67427 | NCBI Gene Expression Omnibus, GSE67427 |

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
