## [Editor Report]

The important work described in this manuscript reveals a new pathway in nutritional immunity: the zinc transporter SLC30A1 in the antimicrobial function of macrophages. Authors provide convincing evidence that zinc homeostasis promotes macrophage cell function that is not conducive to the intracellular proliferation of *Salmonella*, specifically attenuated *Salmonella*. This will be of interest to readers involved in pathogenesis, immunity, and bacteriology.

---

## [Decision Letter]

**Decision letter after peer review:**

Thank you for submitting your article "The zinc transporter Slc30a1 in macrophages plays a protective role against *Salmonella* infection" for consideration by *eLife*. Your article has been reviewed by 3 peer reviewers, and the evaluation has been overseen by a Reviewing Editor and Dominique Soldati-Favre as the Senior Editor.

Essential revisions (for the authors):

1) Check if there is an effect of Slc30A1 deletion on bacterial phagocytosis (evaluation of bacterial numbers after 30 min of infection).

2) How does Zinc or TPEN supplementation to bacteria in LB medium affect the log growth of *Salmonella*?

3) revisit the histopathology results: check the scale bars, quantify lesions and carry out statistical analyses. The current magnification is not informative.

4) Seeing the minor differences in burden presented on Figure 5A with 3 mice, please repeat.

5) revisit the NO involvement in bactericidal activities as per the three reviewer's recommendation.

6) Remove Figure 7.

7) clarify the experimental setup that allow mice to survive such inoculum as per the three reviewers' comments.

*Reviewer #2 (Recommendations for the authors):*

The histopathology needs to be reanalyzed by a pathologist. I don't think the scale bars can be right. 5-10 mm? Quantification of lesions and statistical analyses are needed. The current magnification is not informative.

Many of the experiments include very few numbers. For example, 3 mice were used for data in Figure 4K. This is a very low number of mice.

Figure 5A, the differences in burden at 24 are very minor, and represent an N of 3. This experiment needs to be repeated.

Figure 5B, it is not clear whether the bacteria in the Slec30a1-flox group (right panel) are actually in the vacuole. In this panel, not clear that the top arrow is actually pointing to a bacterial cell as the electrodensity is the particle is quite different from all other bacteria shown.

Figure 5 shows expression of nos2 in response to *Salmonella*. However, panel F and G refer to responses to HK bacteria. The analysis should be done in response to life *Salmonella* cells. Given the MOI of 1 used, I'd be surprised if the macrophages produce much nitrite. Moreover, at the concentrations of nos2 expressed in the experimental conditions used, involvement of NO in the antimicrobial activity of BMDM is limited or inexistent. Killing assays in the presence of a NOS2 inhibitor should be used in order to draw the conclusions made in the paper. The burden isolated from peritoneal macrophages differs at 4 h but not at later times when wild-type and mutant littermates are compared (Figure 4K). The kinetics in Figure 4K are also inconsistent with the idea NO is mediating the resistance associated with Slc30a1. It is quite possible the zinc transport system contributes to NO-mediated host defense, but not under the experimental conditions tested.

As mentioned earlier, the data in Figure 7 is premature, detracting from the main body of work. I suggest data in Figure 7 is deleted from this manuscript.

Published data is not properly discussed, and the present findings are not presented in context of what we know about the role zinc plays in *Salmonella* pathogenesis.

*Salmonella* uses a zinc transport and export systems to gain a hold in mice. Papers on bacterial znuABC and znt transport systems should be considered in the discussion (PMID: 30365536 and PMID: 33203749).

I find it surprising that wild-type mice live so long after infection with 100,000 CFU of *Salmonella*. What is the strain of *Salmonella* used? Does it harbor any attenuating mutations?

*Reviewer #3 (Recommendations for the authors):*

The authors have done a lot of work and the information that Slc30a1 expression in macrophages contributes to control of *Salmonella* infection in mice is a new finding that will be of interest to the field. While the findings that SLC30A1 plays a role in macrophage mediated killing of *Salmonella* appears. The major conclusions of this study are not fully supported by the data and additional experiments are needed to define the mechanism by which Slc30a1 confers protection against bacterial pathogens like *Salmonella* Typhimurium.

Primarily, the stark difference in the survival of control animals compared to Slc30a1fl/flLysMCre mice (Figure 4D) does not appear to be consistent with the modest differences in the bacterial burden in *Salmonella* infected mice (Figure 4K) and macrophages (Figure 5A). The authors should consider reporting additional time points of these experiments or should explain how small differences in the bacterial burden result in the high susceptibility of the Slc30a1 conditional KO animal.

Conventional C57BL/6 mice are already highly susceptible to *Salmonella* infection, however considering the dose administered (1 x 10^5 via IP route), the mice should all die by 5-7 days, yet most of the mice survive for two weeks. The authors should include additional details as to how the animals were derived and the source of the *Salmonella* typhimurium strain used for the study.

*Salmonella* express and secrete various effectors that alter the cellular environment of infected macrophages. As such, it is important that the authors perform additional experiments to show that iNOS and nitrite levels are down in *Salmonella* infected macrophages lacking Slc30a1, as the data shown are with dead *Salmonella* and LPS, and the result with live *Salmonella* may be different. Currently, the data show only a correlation between an increase in *Salmonella* numbers and a decrease in iNOS activity in Slc30a1 deficient macrophages. To infer a mechanistic link between Slc30a1 activity and NO-mediated microbicidal activity additional experiments are required, for example using inhibitors of iNOS.

Overall, there are too many data in the paper, and not all of them are needed to support the main conclusions of the paper. The authors should consider paring down the data shown so that only the key experiments needed to support their conclusions are shown in the main figures and the remaining data are shown in the supplement. A published paper should communicate a scientific advance in a way that is accessible to the reader, and while the authors have clearly done a lot of work, it should not be necessary for the reader to wade through results that are not needed to support the major conclusions of the paper.

[Editors' note: further revisions were suggested prior to acceptance, as described below.]

Thank you for resubmitting your work entitled "The zinc transporter Slc30a1 in macrophages plays a protective role against *Salmonella* infection" for further consideration by *eLife*. Your revised article has been evaluated by Dominique Soldati-Favre (Senior Editor) and a Reviewing Editor.

The manuscript has been improved but there are some remaining minor issues that need to be addressed, as outlined below:

*Reviewer #2 (Recommendations for the authors):*

This much improved manuscript has addressed my previous criticisms. There are two remaining issues that still need addressing. The conditional mice target macrophages but also other cells of the myelocytic cell lineage. Throughout the manuscript, please, soften your claim that the conditional mice used bear a macrophage-specific defect. Second, the fact that the strain of *Salmonella* used is a vaccine explains the high doses tolerated by the mice. The attenuated strain might have been important for the elicitation of NOS2-dependent immunity, a phenotype that may not have been exposed with wild-type *Salmonella*. To be fair with the scientific community, it should be very clear that the studies used an attenuated *Salmonella* strain. Therefore, I recommend the title is changed to "The zinc transporter Slc30a1 in macrophages plays a protective role against attenuated *Salmonella*."

*Reviewer #3 (Recommendations for the authors):*

We appreciate the significant amount of effort made by the authors to carefully address our comments and find the revised manuscript to be much improved. A few editorial changes to the manuscript are needed before publication.

(1) The information on the identity of the attenuated *Salmonella* strain that is provided in the response to reviewers helps to explain why the mice survived for so long after the bacterial challenge. This information still needs to be added to the revised manuscript, in both the Materials and methods and the Results section describing the experiments, to clarify to readers that the experiments were performed with attenuated *Salmonella enterica* subsp. enterica serovar Typhimurium strain MeganVac1 (Accession: CP112994.1).

(2) While the caveat that a mechanistic link between SLC30A1 deficiency and NO production has not been shown in this study is now acknowledged in the discussion, in the abstract the sentence "We demonstrate that Slc30a1 deficient macrophages are defective in intracellular killing due to loss of iNOS and NO activities with consequent inhibition of NF-κB." needs to be modified to say "We demonstrate that Slc30a1 deficient macrophages are defective in intracellular killing, which correlated with reduced activation of NF-κB and reduction in NO production."

---

## [Author Response]

Essential revisions (for the authors):1) Check if there is an effect of Slc30A1 deletion on bacterial phagocytosis (evaluation of bacterial numbers after 30 min of infection).

As advised, we have repeated the experiment and measured the bacterial numbers after 30 min of infection (dashed line in A). The results show that there is no statistical difference in the bacterial numbers after 30 min between *Slc30a1^fl/fl^LysM^Cre^* and *Slc30a1^fl/fl^* BMDMs. Therefore, the reduction of bacterial numbers after 24 hours occurs due to the impairment of intracellular pathogen-killing capacity as the reviewer pointed out.

**Author response image 1. sa2fig1:** (A) Time course of the intracellular pathogen-killing capacity of *Salmonella* infected *Slc30a1^fl/fl^LysM^Cre^* and *Slc30a1^fl/fl^* BMDMs measured in colony-forming units per ml (*n* = 5). (B) Fold change in *Salmonella* survival (CFU/mL) at different time points from A. (C) Representative images of *Salmonella* colonies on solid agar medium at 24 hours. Data are represented as mean ± SEM. *P* values were determined using 2-tailed unpaired Student’s *t-*test. **P*<0.05, ***P*<0.01, and ns, not significant.

2) How does Zinc or TPEN supplementation to bacteria in LB medium affect the log growth of *Salmonella*?

We found that zinc supplementation at both low (20 µM) and high (640 µM) concentrations negatively effects *Salmonella* growth, especially during log phase and stationary phase in the broth culture medium, but not TPEN (20 µM) supplementation. These indicates that high zinc conditions occur at cellular levels such as within phagosomes (Botella et al., 2011) can limit bacterial growth.

**Author response image 2. sa2fig2:** Growth curve (optical density, OD 600 nm) of *Salmonella* in LB medium at different concentrations of ZnSO_4_ and/or TPEN. Bar graph indicating *Salmonella* growth at specific time points. Each value was expressed as mean of triplicates for each testing and data were determined using 2-tailed unpaired Student’s *t-*test. **P*<0.05, ***P*<0.01, ****P*<0.001 and ns, not significant.

3) revisit the histopathology results: check the scale bars, quantify lesions and carry out statistical analyses. The current magnification is not informative.

As advised, we have re-analysed the samples for histological analysis and arrange the figure to make it clear.

4) Seeing the minor differences in burden presented on Figure 5A with 3 mice, please repeat.

We have increased the numbers of the mice into Figure 5A as suggested. (Response to point 1 above).

5) revisit the NO involvement in bactericidal activities as per the three reviewer's recommendation.

As advised, we have added the detailed information related to the NO involvement in bactericidal activities into the revised manuscript. (Response to Reviewer 2, comment 5).

6) Remove Figure 7.

As suggested, the old Figure 7 has been removed in the revision.

7) clarify the experimental setup that allow mice to survive such inoculum as per the three reviewers' comments.

We have added the detailed information of experimental setup including the bacterial inoculum into the revision.

Reviewer #2 (Recommendations for the authors):The histopathology needs to be reanalyzed by a pathologist. I don't think the scale bars can be right. 5-10 mm? Quantification of lesions and statistical analyses are needed. The current magnification is not informative.

As the reviewer suggested, we have re-arranged the figures to make it clear. Histological scoring was done by a pathologist at Zhejiang University who was blinded to the knockout and control groups. Histological changes of spleen (Stranahan et al., 2019) and liver (Thoolen et al., 2010) were scored from 0-4 based on histiocytic inflammation and necrotic foci, respectively.

**Author response table 1. sa2table1:** The tissue sections of spleen were scored as follows.

Lesion	Score	Description
Histiocytic inflammation	0	None
	1	Minimal- filling <25% of marginal zone
	2	Mild- filling 25-50% of marginal zone
	3	Moderate- filling 51-75% of marginal zone
	4	Marked- filling >75% of marginal zone

**Author response table 2. sa2table2:** The tissue sections of liver were scored as follows.

Severity	Proportion of liver affected	Score	Quantifiable finding
None	None	0	None
Marginal or minimal	Very small amount	1	1-2 foci
Slight or few	Small amount	2	3-6 foci
Moderate or several	Medium amount	3	7–12 foci
Marked or many	Large amount	4	>12 foci, coalescing

Reference:

Stranahan LW, Khalaf OH, Garcia-Gonzalez DG, Arenas-Gamboa AM. 2019. Characterization of *Brucella canis* infection in mice. *PLoS One*. 14: e0218809. https://doi: 10.1371/journal.pone.0218809, PMID: 31220185

Thoolen B, Maronpot RR, Harada T, Nyska A, Rousseaux C, Nolte T, Malarkey DE, Kaufmann W, Küttler K, Deschl U, Nakae D, Gregson R, Vinlove MP, Brix AE, Singh B, Belpoggi F, Ward JM. 2010. Proliferative and nonproliferative lesions of the rat and mouse hepatobiliary system. *Toxicologic Pathology.* 38: 5S-81S. https://doi: 10.1177/0192623310386499, PMID: 21191096

Many of the experiments include very few numbers. For example, 3 mice were used for data in Figure 4K. This is a very low number of mice.

The sample size of the present study is relatively small considering the extent of different time points. In addition, the mice numbers at 48 hpi were decreased due the incubation period of *Salmonella* infection. Some mice that are susceptible to *Salmonella* could not survive.

However, we agree that 3 mice for the data in Figure 4K (now Figure 4G) is a very low number. Therefore, we have repeated the experiment by adding up to 7 mice per group after 48 hpi into Figure 4K (now Figure 4G).

**Author response image 3. sa2fig3:** Bar graph indicates fold change in *Salmonella* burden of *Slc30a1^fl/fl^LysM^Cre^* normalized to *Slc30a1^fl/fl^*. Data are represented as mean ± SEM. *P* values were determined using 2-tailed unpaired Student’s *t-*test. **P*<0.05, ****P*<0.001, and ns, not significant.

Figure 5A, the differences in burden at 24 are very minor, and represent an N of 3. This experiment needs to be repeated.

As the reviewer suggested, we have repeated the experiment by using BMDMs from 5 mice. Fold change in *Salmonella* survival is presented here as a bar graph to compare the different between both groups (Figure 5A).

Figure 5B, it is not clear whether the bacteria in the Slec30a1-flox group (right panel) are actually in the vacuole. In this panel, not clear that the top arrow is actually pointing to a bacterial cell as the electrodensity is the particle is quite different from all other bacteria shown.

As the reviewer suggestion, we have re-arranged the figure to make it clear (Figure 5B).

Figure 5 shows expression of nos2 in response to *Salmonella*. However, panel F and G refer to responses to HK bacteria. The analysis should be done in response to life Salmonella cells.

As the reviewer suggested, we have added the data into the revise manuscript (Figure 5F, G) and moved the data of HK-induced macrophages to the revised supplementary (Figure supplement 6C and E).

Given the MOI of 1 used, I'd be surprised if the macrophages produce much nitrite. Moreover, at the concentrations of nos2 expressed in the experimental conditions used, involvement of NO in the antimicrobial activity of BMDM is limited or inexistent.

Since a nitric oxide boost takes time, macrophages were stimulated with HK-*Salmonella* cells at MOI = 10 for 24 hours. Adding life *Salmonella* cells directly can cause the bacterial overgrowth in the cell culture medium as time passes, especially after log phage of bacterial growth. However, according to the above comment, we agree with the reviewer that the macrophages should be done in response to life *Salmonella* cells. We therefore set upped the experiment based on bacterial killing assay which extracellular bacterial growth is prevented, and then detected nitrite concentrations in the cell supernatant.

Killing assays in the presence of a NOS2 inhibitor should be used in order to draw the conclusions made in the paper.

We thank for this tempting suggestion. We agree that it would draw the clear conclusion in the paper. However, Slc30a1 might regulate NO at different levels as it is a cause of high intracellular free zinc and other zinc binding proteins like Mt1. Therefore, it is difficult to make a conclusion which inhibitors are needed to be applied. However, the reviewer has raised an important point of mechanistic study that needed to be explored further. Accordingly, we have added this as a limitation on the revised manuscript (Page 12, Line 307-314)

The burden isolated from peritoneal macrophages differs at 4 h but not at later times when wild-type and mutant littermates are compared (Figure 4K).

We found that *Salmonella* burden in peritoneal fluid in Figure 4K (now Figure 4G) seems to be consistent with the proportion of peritoneal macrophages in Figure 4I (now Figure 4F). We suppose that Slc30a1 deletion probably attenuates the macrophage functions and/or delays cell recruitments, thereby increases the numbers of *Salmonella* burden at the early period of infection but not at later times. Many previous studies reported that defects in peritoneal macrophage recruitment increase mortality and bacterial loads in mice (Jorch et al., 2019, Leendertse et al., 2009, Zhang et al., 2019).

Reference:

Jorch SK, Surewaard BG, Hossain M, Peiseler M, Deppermann C, Deng J, Bogoslowski A, van der Wal F, Omri A, Hickey MJ, Kubes P. 2019. Peritoneal GATA6+ macrophages function as a portal for *Staphylococcus aureus* dissemination. *The Journal of Clinical Investigation.* 129: 4643-4656. https://doi: 10.1172/JCI127286, PMID: 31545300

Leendertse M, Willems RJ, Giebelen IA, Roelofs JJ, van Rooijen N, Bonten MJ, van der Poll T. 2009. Peritoneal macrophages are important for the early containment of Enterococcus faecium peritonitis in mice. *Innate Immunity*. 15:3-12. https://doi: 10.1177/1753425908100238, PMID: 19201820

Zhang N, Czepielewski RS, Jarjour NN, Erlich EC, Esaulova E, Saunders BT, Grover SP, Cleuren AC, Broze GJ, Edelson BT, Mackman N, Zinselmeyer BH, Randolph GJ. 2019. Expression of factor V by resident macrophages boosts host defense in the peritoneal cavity. *Journal of Experimental Medicine.* 216: 1291-1300. https://doi: 10.1084/jem.20182024, PMID: 31048328

The kinetics in Figure 4K are also inconsistent with the idea NO is mediating the resistance associated with Slc30a1. It is quite possible the zinc transport system contributes to NO-mediated host defense, but not under the experimental conditions tested.

We thank the reviewer for this constructive suggestion. In response, we have observed the expression of *nos2* in the spleen, liver, and peritoneal macrophages of *Slc30a1^fl/fl^LysM^Cre^* mice in comparison with *Slc30a1^fl/fl^* mice upon *Salmonella* infection. We found that the *nos2* expression of spleen in *Slc30a1^fl/fl^LysM^Cre^* mice were lower than that of the control group but not in liver. These might be supported by increased zinc accumulation and Mt1 overexpression in the spleen and BMDMs as shown in Figure 6.

**Author response image 4. sa2fig4:** RT-qPCR analysis of mRNA encoding *Nos2* in the spleen, liver, and peritoneal macrophages of uninfected and *Salmonella* infected mice for 24 h. Data are represented as mean ± SEM. *P* values were determined using 2-tailed unpaired Student’s *t-*test, **P*<0.05, ***P*<0.01 and ns, not significant.

As mentioned earlier, the data in Figure 7 is premature, detracting from the main body of work. I suggest data in Figure 7 is deleted from this manuscript.

We agree with the reviewer’s suggestion. We have accordingly removed Figure 7 from the revised manuscript.

Published data is not properly discussed, and the present findings are not presented in context of what we know about the role zinc plays in *Salmonella* pathogenesis.*Salmonella* uses a zinc transport and export systems to gain a hold in mice. Papers on bacterial znuABC and znt transport systems should be considered in the discussion (PMID: 30365536 and PMID: 33203749).

As the reviewer suggested, the conclusion section has been rewritten to make it clear (Page 12, Line 315-320). In addition, we have added the recommended articles to support our discussion (Page 12, Line 293-296).

I find it surprising that wild-type mice live so long after infection with 100,000 CFU of *Salmonella*. What is the strain of *Salmonella* used? Does it harbor any attenuating mutations?

The *Salmonella* stain is a gift from our friend, Professor Ge Bao-xue. We have sent this stain for genetic characterisation which we found 100% identity to *Salmonella enterica* Typhimurium with many strains originated from poultry. One of them is *Salmonella enterica* subsp. *enterica* serovar Typhimurium strain MeganVac1 (Accession: CP112994.1), a live attenuated stain. We suppose that this would support the relationship between the high infectious dose and mice survive.

Reviewer #3 (Recommendations for the authors):The authors have done a lot of work and the information that Slc30a1 expression in macrophages contributes to control of *Salmonella* infection in mice is a new finding that will be of interest to the field. While the findings that SLC30A1 plays a role in macrophage mediated killing of *Salmonella* appears. The major conclusions of this study are not fully supported by the data and additional experiments are needed to define the mechanism by which Slc30a1 confers protection against bacterial pathogens like *Salmonella* Typhimurium.

We thank the reviewer for this constructive suggestion. We clarify in the conclusion section accordingly (Page 12, Line 315–320)

Primarily, the stark difference in the survival of control animals compared to Slc30a1fl/flLysMCre mice (Figure 4D) does not appear to be consistent with the modest differences in the bacterial burden in *Salmonella* infected mice (Figure 4K) and macrophages (Figure 5A). The authors should consider reporting additional time points of these experiments or should explain how small differences in the bacterial burden result in the high susceptibility of the Slc30a1 conditional KO animal.

As encouraged, we have re-examined the survival rate with a greater number of animals to make the data clear in Figure 4D (now Figure 4C). In addition, we have increased the sample size of the mice in Figure 4K (now Figure 4G) and repeated the experiment in Figure 5A to confirm the defect of bacterial clearance by macrophages. To address the differences in the bacterial burden, we have provided the bar graphs indicating the fold change of *Salmonella* burden of KO normalized to WT animals. We agree that it would help explain the difference in the bacterial burden. However, due to time-consuming, expensive of animal tests and handling of pathogenic microorganisms, we hope the reviewer would agree, considering the additional points for further study.

**Author response image 5. sa2fig5:** Bar graph indicates fold change in *Salmonella* survival (CFU/mL) (*Right*). *P* values were determined using 2-tailed unpaired Student’s *t-*test. **P*<0.05, ***P*<0.01, ****P*<0.001, and ns, not significant.

Conventional C57BL/6 mice are already highly susceptible to *Salmonella* infection, however considering the dose administered (1 x 10^5 via IP route), the mice should all die by 5-7 days, yet most of the mice survive for two weeks. The authors should include additional details as to how the animals were derived and the source of the *Salmonella* typhimurium strain used for the study.

As the reviewer suggested, we have added the detailed genotype of *Slc30a1^fl/fl^LysM^Cre^* mice to the revised supplementary (Figure supplement 10). The *Salmonella* stain is a gift from our friend, Professor Ge Bao-xue. We have sent this stain for genetic characterisation which we found 100% identity to *Salmonella enterica* Typhimurium with many strains originated from poultry. One of them is *Salmonella enterica* subsp. *enterica* serovar Typhimurium strain MeganVac1 (Accession: CP112994.1), a live attenuated stain. We hope that this would support the relationship between the high infectious dose and mice survive.

*Salmonella* express and secrete various effectors that alter the cellular environment of infected macrophages. As such, it is important that the authors perform additional experiments to show that iNOS and nitrite levels are down in *Salmonella* infected macrophages lacking Slc30a1, as the data shown are with dead *Salmonella* and LPS, and the result with live *Salmonella* may be different.

As the reviewer suggested, we have added the result with live *Salmonella* in the revised manuscript (Figure 5F, G).

Currently, the data show only a correlation between an increase in *Salmonella* numbers and a decrease in iNOS activity in Slc30a1 deficient macrophages. To infer a mechanistic link between Slc30a1 activity and NO-mediated microbicidal activity additional experiments are required, for example using inhibitors of iNOS.

We agree with the reviewer’s suggestion that the data show only a correlation between an increase in *Salmonella* numbers and a decrease in iNOS activity in Slc30a1 deficient macrophages. However, this is the first study of Slc30a1 in regulation of iNOS/NO in macrophages. We suppose that absence of Slc30a1 might impair iNOS/NO activity by multiple mechanisms and more research is needed to further address the precise mechanism using iNOS inhibitors: we have added these as a limitation on the revised manuscript (Page 12, Line 307-314)

Overall, there are too many data in the paper, and not all of them are needed to support the main conclusions of the paper. The authors should consider paring down the data shown so that only the key experiments needed to support their conclusions are shown in the main figures and the remaining data are shown in the supplement. A published paper should communicate a scientific advance in a way that is accessible to the reader, and while the authors have clearly done a lot of work, it should not be necessary for the reader to wade through results that are not needed to support the major conclusions of the paper.

We appreciate the reviewer for this constructive suggestion. In response, we have carefully arranged the results and shown the key experiments in the main figures. We hope that these revisions and the improved text will be satisfactory. A detailed information for all changes is provided below.

Figure 1: Schematic image Figure 1A and 1H have been eliminated. Results in Figure 1G and 1H have been removed to the revised supplementary (Figure supplement 2A, B).

Figure 2: Schematic image Figure 2B and 2E have been eliminated.

Figure 3: Results in Figure 3F and 3G have been removed to the revised supplementary (Figure supplement 3A, B).

Figure 4: Blood parameters including serum TNFa (Figure 4F), serum ALT (Figure 4G), serum AST (Figure 4H), numbers of neutrophiles and macrophages (Figure 4J) have been removed to the revised supplementary (Figure supplement 4 and Table supplement 3).

Figure 5: Schematic image Figure 5A has been eliminated.

Figure 6: Schematic image Figure 6A, 6D, and 6H have been eliminated. Results in Figure 6E has been removed to the revised supplementary (Figure supplement 8).

Figure 7: We have eliminated old Figure 7 and refined the entire manuscript.

[Editors’ note: what follows is the authors’ response to the second round of review.]

The manuscript has been improved but there are some remaining minor issues that need to be addressed, as outlined below:

We thank the reviewer for your constructive suggestions. In the revised manuscript, we have carefully addressed your comments and hope the revision meets your expectations.

Reviewer #2 (Recommendations for the authors):This much improved manuscript has addressed my previous criticisms. There are two remaining issues that still need addressing. The conditional mice target macrophages but also other cells of the myelocytic cell lineage. Throughout the manuscript, please, soften your claim that the conditional mice used bear a macrophage-specific defect.

As suggested, we have changed “macrophage-specific *Slc30a1* knockout mice” to “*Lyz2*-Cre driven *Slc30a1* conditional knockout mice” throughout the entire manuscript (Page 2, line 26; page 6, line 146; page 9, line 235; page 11, line 209, 304-305, 314; page 13, line 361-362; page 29, line 813, 828; Page 31, line 877).

Second, the fact that the strain of *Salmonella* used is a vaccine explains the high doses tolerated by the mice. The attenuated strain might have been important for the elicitation of NOS2-dependent immunity, a phenotype that may not have been exposed with wild-type *Salmonella*. To be fair with the scientific community, it should be very clear that the studies used an attenuated *Salmonella* strain. Therefore, I recommend the title is changed to "The zinc transporter Slc30a1 in macrophages plays a protective role against attenuated *Salmonella*."

This is a very good point. As you advised, we have changed the title to "The zinc transporter Slc30a1 in macrophages plays a protective role against attenuated *Salmonella*."

Reviewer #3 (Recommendations for the authors):We appreciate the significant amount of effort made by the authors to carefully address our comments and find the revised manuscript to be much improved. A few editorial changes to the manuscript are needed before publication.

We thank the reviewer for your positive comments

(1) The information on the identity of the attenuated *Salmonella* strain that is provided in the response to reviewers helps to explain why the mice survived for so long after the bacterial challenge. This information still needs to be added to the revised manuscript, in both the Materials and methods and the Results section describing the experiments, to clarify to readers that the experiments were performed with attenuated *Salmonella* enterica subsp. enterica serovar Typhimurium strain MeganVac1 (Accession: CP112994.1).

As the reviewer advised, we have added the related information of the identity of the attenuated *Salmonella* stain to the section of the Materials and methods (Page 14, line 383384), Abstract (Page, line 20-21) and Results sections (Page 4, line 84-85; page 5, line 110, 114, 124; page 6, line 136, 155; page 7, line 179, 188; page 8, line 200; page 9, line 238).

(2) While the caveat that a mechanistic link between SLC30A1 deficiency and NO production has not been shown in this study is now acknowledged in the discussion, in the abstract the sentence "We demonstrate that Slc30a1 deficient macrophages are defective in intracellular killing due to loss of iNOS and NO activities with consequent inhibition of NF-κB." needs to be modified to say "We demonstrate that Slc30a1 deficient macrophages are defective in intracellular killing, which correlated with reduced activation of NF-κB and reduction in NO production."

Thank you very much for this great suggestion. We have modified the sentence in the abstract as the reviewer suggested (Page 2, line 28-29).